# The T-Cell Response to Type 2 Porcine Reproductive and Respiratory Syndrome Virus (PRRSV)

**DOI:** 10.3390/v11090796

**Published:** 2019-08-29

**Authors:** Andrew R. Kick, Amanda F. Amaral, Lizette M. Cortes, Jonathan E. Fogle, Elisa Crisci, Glen W. Almond, Tobias Käser

**Affiliations:** Department of Population Health and Pathobiology, College of Veterinary Medicine, North Carolina State University, Raleigh, NC 27607, USA

**Keywords:** PRRSV, T cells, adaptive immunity, treg, thelper, Cytotoxic proliferation, TCR-γδ, IFN-γ

## Abstract

Porcine reproductive and respiratory syndrome virus (PRRSV) continues to cause severe reproductive and respiratory pathologies resulting in immense monetary and welfare costs for the swine industry. The vaccines against PRRSV are available; but they struggle with providing protection against the plethora of heterologous PRRSV strains. To improve PRRSV vaccine development, the aim of this study was to provide an in-depth analysis of the crucial heterologous T-cell response to type-2 PRRSV. Following PRRSV modified live virus (MLV) vaccination or infection using one high- or one low-pathogenic PRRSV-strain, this nine-week study evaluated the T-cell response to different PRRSV strains. Our results demonstrate an important role for T cells in this homo- and heterologous response. Specifically, the T-helper cells were the main responders during viremia. Their peak response at 28 dpi correlated with a reduction in viremia, and their homing receptor expression indicated the additional importance for the anti-PRRSV response in the lymphatic and lung tissue. The cytotoxic T lymphocyte (CTL) response was the strongest at the site of infection—the lung and bronchoalveolar lavage. The TCR-γδ T cells were the main responders post viremia and PRRSV induced their expression of the lymph node homing the chemokine receptor, CCR7: This indicates a crucial role for TCR-γδ T cells in the anti-PRRSV response in the lymphatic system.

## 1. Introduction

Porcine reproductive and respiratory syndrome virus Type 2 (PRRSV-2) causes severe reproductive and respiratory pathology leading to tremendous financial losses for the swine industry. PRRSV-2 is often involved in the porcine respiratory disease complex (PRDC) decreasing health and resulting in the reduced performance, higher medication costs and elevated mortality [1,2]. One main characteristic of PRRSV-2 is its ability to actively suppress the host immune response facilitating prolonged viremia, persistence in tissues, and the development of chronic secondary infections associated with PRDC. The following mechanisms of PRRSV-2 immune suppression were reported: The inhibition of NK cell activity and Type I Interferons production in the innate response [3]; the induction of PRRSV-2 specific CD4^+^CD25^+^ regulatory T cells (Tregs) [4,5,6,7,8,9,10,11,12]; and the dysregulation of T-cell development in the neonatal thymus [13]. Ideally for the host, an effective T-cell response to PRRSV-2 would display the characteristics of a strong T-helper type 1 (Th1) response: Early CD4^+^ Th proliferation and significant production of IFN-γ; the differentiation of PRRSV-reactive Th cells into both, lymph node homing central memory (T_CM_) and tissue homing effector memory (T_EM_) populations [14]; and the induction of multifunctional memory T cells (T_multi_). These T_multi_ combine the best characteristics of memory cells: T_multi_ have a long lifespan; they produce high levels of cytokines, such as IFN-γ, IL-2, and TNF-α; they can home to the lymph nodes and tissue; they have been identified as the best correlates of protection for many viral diseases [15]. In addition, the early activation and proliferation of cytotoxic T cells are crucial in the response to viral infections [16]. To facilitate the clearance of the virus, the induction of immunosuppressive Tregs should occur towards the end of the infection. The role and function of TCR-γδ T cells in an ideal immune response to PRRSV-2 is unclear with limited research on the topic, but they possibly serve as an additional stimulatory and regulatory link between the innate and adaptive immunity [17,18]. The modified live virus (MLV) vaccines have been licensed in the US for 25 years. It was demonstrated that these vaccines provide protection in vaccinated animals against homologous strains and limited protection against heterologous strains. However, commercially available MLV vaccines have not provided complete protection from PRRSV-2 outbreaks, possibly due to the PRRSV-2 high mutation rate and strain diversity [19]. In order to provide efficient protection, vaccines must induce long-lasting immunological memory against multiple heterologous PRRSV-2 strains.

Despite the importance of PRRSV-2 and its progress in commercially available vaccines, there remains a large gap in the knowledge of the adaptive immune response over the course of the infection and the clearance of viremia [1]. This paper addresses four specific gaps identified by Loving et al. and Lunney et al.: (1) To develop a methodology to investigate the antigen-specific expansion of the T-cell subsets and the establishment of memory cell populations [1]; (2) to develop a system to evaluate the PRRSV-2 vaccine induced cross-protection to the heterologous virus strain [2]; (3) to investigate the time course of Treg induction during infection [1]; and (4) the characterization of IFN-γ producing T-cell subsets [1]. The present study accomplishes those objectives by systematically examining the porcine T-cell response to three different PRRSV-2 strains: This study i) vaccinated weanlings using a commercially available modified live virus (MLV) vaccine, or ii) infected them with PRRSV-2 strains currently circulating in North Carolina with either a low-pathogenic (LP) 1-3-4 strain, or a high-pathogenic (HP) 1-7-4 strain. This study then followed the infection and the induced T-cell immune response for nine weeks. The specific T-cell response to the homologous and heterologous PRRSV-2 strains was studied by multi-color flow cytometry upon the in vitro re-stimulation of isolated immune cells with these three virus strains.

In this study, the immune response of T cells to homologous and heterologous PRRSV-2 strains are summarized: The strain-specific T-cell subset proliferation, cytokine production, and differentiation into memory cells are determined; the peak of the Th response is identified 28 days post infection; and that response is able to be correlated with the clearance of viremia. Thereby, this study addresses critical knowledge gaps in the immune response to PRRSV-2 and it represents the most detailed analysis of the T-cell response to PRRSV-2 to date.

## 2. Materials and Methods

### 2.1. PRRSV-2 Propagation and Titration

The PRRSV-2 viral isolates (strains LP (1-3-4) and HP (1-7-4)) were obtained from a commercial NC pork company. The strains were verified with Open Reading Frame (ORF) 5 sequencing and the restriction fragment length polymorphism (RFLP) was performed by Iowa State University Veterinary Diagnostic Laboratory (Ames, IA, USA). For the inoculation, both PRRSV-2 strains were passed three times through PRRSV-negative porcine alveolar macrophages (PAMs) as previously described [9] with the following modifications. RPMI-1640, 1X with L-glutamine (Corning, Corning, NY, USA) supplemented with 10% fetal bovine serum (FBS) (VWR, Radnor, PA, USA) and 1X penicillin/streptomycin (Corning) was used as media. For the third passage, once the cytopathic effect was observed (approximately 30 h), T-75 flasks (Sarstedt, Nümbrecht, Germany) were freeze thawed one time and a supernatant was spun at 650× *g* at 4 °C for 20 min. To concentrate the virus, the supernatant was spun in a Vivaspin 20, 30kDA MWCO (GE Healthcare, Buckinghamshire, UK) at 8000× *g* at 4 °C for 45 min. The viral concentrate was then sterile filtered using a 0.22 µm filter (Corning) and stored in aliquots at −80 °C. The supernatant from the cells with the same treatment, except viral infection, were used as the control (mock).

For ex vivo viral restimulation, the three strains (LP, HP, and a commercially available modified live virus (MLV) vaccine) were propagated separately in a 500-mL spinner flask (Corning) on MA-104 cells (ATCC, Manassas, VA, USA) adapted from a previously described method [20]. Briefly, 500-mL Minimum Essential Medium Eagle (MEM, 1X) (Corning) supplemented with 10% FBS (VWR) and 1X penicillin/streptomycin (Corning) was added to the flask along with 1 g of Corning Enhanced Attachment Microcarrier beads (Corning, Kennebunk, ME, USA) rehydrated in 50-mL sterile 1X PBS. The flask was placed on a magnetic stirrer in an incubator at 37 °C and 5% CO_2_. Approximately 4 × 10^7^ MA-104 cells grown in T-75 flasks (Sarstedt) were added to the spinner flask and incubated at a minimal stirring rate for approximately 4 days until the bead samples were determined to be confluent with the cells by light microscopy. The respective virus stock was added to achieve an approximate 10^4^ 50% tissue culture infectious dose (TCID_50_)/ml. After 4 days, the MA-104 cells in the culture displayed a cytopathic effect of approximately 80%. The supernatant was transferred to 50-mL flasks and frozen at −20 °C. The supernatant was thawed and spun at 2200× *g* at 4 °C for 10 min. The supernatant was then transferred to 36 mL Nalgene centrifuge tubes (Thermo Fisher Scientific, Rochester, NY, USA) and spun in a Sorvall 100S Ultracentrifuge (Sorvall (Thermo Fisher Scientific), Newtown, CT, USA) at 73,000× *g* at 4 °C for 2 h. The supernatant was discarded and the pellet was resuspended in media (MEM complete), sterile filtered, and stored in 100 µL aliquots at −80 °C.

The TCID_50_ titers of viral stock solutions were determined utilizing PAMs for viral inoculation and MA-104 cells for viral restimulation as previously described (Spearman-Karber TCID50 method according to the OIE manual of diagnostic tests (OIE, “Chapter 2.8.7 Porcine Reproductive and Respiratory Syndrome,” Terr. Man., no. May 2015, 2015).

### 2.2. Study Design

Twenty-four 4-week-old weaned piglets from a PRRSV-negative herd (NC State University Swine Education Unit, Raleigh, NC, USA) were moved to the BSL-2 Laboratory Animal Research (LAR) facility at NC State University, College of Veterinary Medicine (Raleigh, NC, USA). The pigs were randomly assigned by sow and sex into one of four treatment groups with six pigs each (3 gilts/3 barrows). The assigned treatment groups were the control (MOCK), vaccinated with a commercially available MLV (MLV), inoculated with NC PRRSV-2 strain 1-3-4 (LP), or inoculated with NC PRRSV-2 strain 1-7-4 (HP) (Figure 1A). Each group was placed in a separate room with three pigs/pen. The pigs were fed and monitored two times/day in accordance with the USDA recommendations. For inoculation, the pigs were manually restrained. The MOCK pigs received a 1 mL dose of PAM culture media injected intranasally (IN) (500 µL per nostril). The MLV pigs received a 2 mL intramuscular (IM) injection of the commercially available vaccine in accordance with the manufacturer’s instructions. The LP and HP pigs received a 10^6^ TCID_50_/_mL_ dose of the virus injected intranasally (500 µL/nostril). The blood samples, body weight, and rectal temperature were collected weekly for nine weeks post infection (Figure 1A). At nine weeks post infection (64–67 dpi—one treatment group per day), the animals were sacrificed using lethal injection. Then, the spleen, tonsils, mediastinal and tracheobronchial lymph nodes, and lungs were harvested. The experimental procedures were approved by the NC State University Institutional Animal Care and Use Committee (IACUC) ID# 17-166A (29. Nov., 2017).

### 2.3. Blood and Tissue Sample Processing

Sample processing was conducted as previously described [21]. Briefly, for serum collection, the blood was collected into SST tubes (BD Biosciences, San Jose, CA, USA), incubated for two hours, spun at 2000× *g* for 20 min at 23 °C and stored in aliquots at −80 °C. The complete blood counts (CBCs) were determined from the blood collected in EDTA tubes (BD Biosciences) using a Hemavet 950 system (Drew Scientific Group, Miami Lakes, FL, USA). The blood for peripheral blood mononuclear cells (PBMC) isolation was collected in Heparin tubes (BD Biosciences) and performed by density centrifugation using Ficoll-Paque (GE Healthcare, Uppsala, Sweden). After isolation, the PBMCs were frozen in 10% DMSO, 40% FBS, and 50% RPMI-1640 and stored in liquid nitrogen until they were thawed for laboratory analysis.

At necropsy, the tissue samples were minced, pressed through a tea sieve, flushed with ice-cold PBS and passed through a 40 µm cell strainer (BD Biosciences) to obtain the single cell suspensions. The cell suspension was then layered onto Ficoll-Paque (GE Healthcare) in order to obtain mononuclear cells. The bronchial alveolar lavage (BAL) was performed as previously described [22]. The isolated BAL cells were saved for immunotyping or were frozen in 10% DMSO, 40% FBS, and 50% RPMI-1640 and stored in liquid nitrogen until they were thawed for laboratory analysis.

### 2.4. Immunotyping of Tissue Samples

All flow cytometry (FCM) stainings and protocols are adapted from those previously established [21]. Single-cell suspensions from the spleen, tonsils, mediastinal lymph nodes (MS LN), tracheobronchial lymph nodes (TB LN), lungs and BAL were stained for T-cell subset and innate immune subset discrimination markers, as well as, for PRRSV-2 infection. The FCM staining panels and gating hierarchies are summarized in Appendix A. The gating strategy for myeloid cells in the PRRSV staining panel is based upon those previously established [23]. The cells were recorded on a Cytoflex using the CytExpert software (Beckman Coulter, Brea, CA, USA). The data analysis was performed with FlowJo version 10.5.3 (FLOWJO LLC, Ashland, OR, USA) with gates based upon the fluorescence minus one (FMO) controls.

### 2.5. Cytokine Production of PRRSV-Stimulated T-cell Subsets

The frozen PBMCs were thawed and seeded at a density of 500,000 PBMCs per well in quadruplicate in RPMI-1640 supplemented with 10% FBS and 1X antibiotic-antimycotic (Corning). The cells were stimulated with one of four treatments: Media, MLV, LP, or HP PRRSV virus (MOI of 0.1). Phorbol 12-myristate 13-acetate (PMA, 5 ng/mL, Alfa Aesar, Ward Hill, MA, USA)/Ionomycin (500 ng/mL, AdipoGen, San Diego, CA, USA) stimulation served as the positive control. The plates were cultured for 18 h and Monensin (5 µg/mL, Alfa Aesar) was added for the last 4 h of culture. The replicates were then pooled and stained in accordance with Appendix A. The cells were recorded on a Cytoflex using the CytExpert software (Beckman Coulter). The data analysis was performed with FlowJo version 10.5.3 (FLOWJO LLC) with gates based upon the FMO controls.

### 2.6. Proliferation of PRRSV-Stimulated T-cell Subsets

While the PBMCs were thawed from liquid nitrogen, the single cell suspensions from the tracheobronchial lymph nodes were used fresh. Both were stained with CellTrace^TM^ Violet Cell Proliferation Kit (Invitrogen) according to the manufacturers’ instructions. Then, the stained cells were seeded into a 96-well round-bottom plate (Sarstedt): Quadruplicates of 200,000 PBMC per well were cultured in RPMI-1640 supplemented with 10% FBS and 1X antibiotic-antimycotic (Corning). The cells were stimulated with one of four treatments: Media, MLV, LP, HP. Concanavalin A (ConA, 2.5 µg/mL, Alfa Aesar) stimulated cells were used as the positive control. The three PRRSV-2 strains were added at a MOI of 0.1. The plates were cultured for 90 h. The replicates were then pooled and stained in accordance with Appendix A for PBMCs and Appendix A for tracheobronchial lymph nodes cell suspensions. The PBMCs were recorded on a Cytoflex using the CytExpert software (Beckman Coulter). The lymph node cells suspensions were recorded on a LSR II (BD Biosciences, San Jose, CA, USA) using FACSDIVA (BD Biosciences) software. All data analysis was performed with FlowJo version 10.5.3 (FLOWJO LLC) with gates based upon the FMO controls.

### 2.7. Viremia

The isolated serum was shipped to Iowa State University Veterinary Diagnostic Laboratory (Ames, IA, USA). Their laboratory determined viremia using quantitative PCR on these serum samples. The results were provided as Ct values and genomic copy numbers/mL.

### 2.8. Statistical Analysis

The statistical analyses were performed using Graphpad Prism 8 (Graphpad Software, San Diego, CA, USA). The data throughout the study were analyzed using a repeated-measure two-way ANOVA. Depending upon the figure, the two factors varied between days post infection (dpi), in vivo inoculation, or restimulation. The significant differences were determined by Fisher’s least significant difference (LSD) and are annotated on the figures. For Figures 2, 3 and 5–9, the box and whisker plots are utilized to display the data. In the box and whisker plots, the line designates the median. The box represents the 25th and 75th percentile and the whiskers designate the smallest and largest value. For Figure 4, the correlation matrix application was utilized with Pearson correlation coefficients and a two-tailed 95% confidence interval. A nonlinear regression curve was fitted utilizing a lognormal equation.

## 3. Results

### 3.1. Clinical Measures, and Viremia

The study design is described in the materials and methods and it is illustrated in Figure 1A. For clarity throughout the results and discussion, the treatment groups are referred to by their in vivo inoculation: MOCK (media), MLV (MLV vaccine), LP (1-3-4), and HP (1-7-4). Within the text, the inoculated animals/pigs refer to the MLV, LP, and HP treatment groups. The infected animals/pigs refer to only the LP and HP treatment groups. Several clinical measures were collected and analyzed over the course of the study and describe the PRRSV-2 infection and symptoms. Viremia is depicted in Figure 1B: Viremia peaked between 7–14 days post inoculation (dpi); and it decreased rapidly in the infected pigs between 21 and 35 dpi. At 7 and 21 dpi, viremia in the HP pigs was significantly higher compared to the LP and MLV pigs. In contrast, at 14 dpi, the LP pigs had a significantly higher viremia. Viremia in the LP and HP pigs was cleared by 49 dpi and 35 dpi respectively, with a re-bounce at 56 dpi. In contrast, MLV pig viremia continued at low levels to 63 dpi (Figure 1B). Peripheral blood lymphocyte (PBL) counts are depicted in Figure 1C: PBLs increased significantly beginning at 21 dpi in LP pigs; both infected groups were significantly higher than MOCK animals at 28 dpi. In general, PBLs from the LP and HP pigs remained significantly higher than MOCK pigs through 63 dpi (Figure 1C). The body weight changes are depicted in Figure 1D: The body weight from infected pigs trended below the MLV and MOCK animals; the body weight from the HP pigs was significantly lower than the MOCK pigs at 49 and 63 dpi (Figure 1D). The body temperatures, which showed a relatively strong variability, are depicted in Figure 1E: With some by day exceptions, the significant differences in body temperature began at 3 dpi in the infected pigs and persisted out to 35 dpi in the HP pigs (Figure 1E). General clinical monitoring revealed that in contrast to the MOCK and MLV pigs, both infected groups exhibited lethargy and a lack of appetite up to 28 dpi. The LP pigs displayed the most pronounced symptoms of infection including respiratory distress and sneezing until 21 dpi. These data confirm that all and only the PRRSV-inoculated pigs developed active infections, while the MOCK and MLV pigs remained healthy, and both infected groups developed symptoms consistent with the established PRRSV-2 pathology.

### 3.2. Time Kinetic of the Systemic Proliferative T-cell Response to Homologous Restimulation

The proliferative response of T-cell subsets to homologous PRRSV-2 strains is depicted in Figure 2. The significance of differences was analyzed within a group comparing the values of each day to 0 dpi. While the MOCK pigs did not respond to PRRSV-2 restimulation (MLV strain), all PRRSV-inoculated groups developed a homologous T-cell response: The MLV and HP pigs proliferated the strongest at 28 dpi; the LP pigs showed the highest response at 42 dpi. These time points also represent the last time points before viremia was cleared in these groups (Figure 1B). The T-cell proliferation in the LP and HP infected pigs remained significantly higher than 0 dpi through 56 dpi (Figure 2A).

The proliferation of all TCR-αβ subsets followed the above described kinetic (Figure 2B: Th cells, C: Tregs, E: CTLs [CD3^+^CD4^−^TCR-γδ^−^CD8α^+^]). Within these TCR-αβ cells, Th cells showed by far the strongest proliferative response. These Th cells are crucial regulators of the immune response. The balance of activated Th cells and Tregs can shift this response either into an inflammatory or immunosuppressive direction. To get an impression of this balance, Figure 2D shows the ratio between proliferating the Th cells and Tregs. The MOCK pigs did not show any differences in this balance. While the MLV pigs showed a decrease in this ratio at 14 and 56 dpi, these values are comparable to the mean of all 0 dpi and the MOCK values. These values can be used as a baseline instead of the highly variable MLV 0 dpi value. Using this baseline for the MLV pigs, all PRRSV-2 inoculated groups showed the same pattern: At an early time-point (14 dpi) and after viremia (56 dpi), the Th/Treg ratio displayed an unaltered baseline level. In contrast, at 28 and 42 dpi, all PRRSV-2 inoculated groups had an increased Th/Treg ratio, indicating an inflammatory systemic immune response environment. However, this ratio never dropped into a significantly low Th/Treg ratio.

Compared to TCR-αβ T cells, the proliferative response of TCR-γδ T cells shifted towards the later time points: While Th cells dominated the response at 14 and 28 dpi, TCR-γδ T cells showed the strongest proliferation to all homologous PRRSV-2 restimulations at 42 and 56 dpi. This proliferation peaked at the latest time point of analysis—56 dpi and peaked after clearance of the initial viremia in LP and HP pigs at 56 dpi.

Overall, these data indicate that Th cells, Tregs, CTLs and TCR-γδ T cells are all involved in the T-cell response to PRRSV-2. The Th cells exhibit the strongest proliferation to homologous viral restimulation during viremia, while TCR-γδ T-cell proliferation comprises the greatest percentage of T cells post-viremia. The heterologous response mimics the homologous response timeline; but the degree of the response varies for the different heterologous strains. Due to the complexity of the data, the following figures focus on the T-cell response to homologous and heterologous PRRSV-2 strains at 28 dpi. This time point was selected for two reasons: i) Most responses peaked at that time point; and ii) PRRSV-2 viremia dropped right after that time point indicating the occurrence of an effective immune response at 28 dpi.

### 3.3. CD4^+^ T-Cell Immune Response to Homo- and Heterologous PRRSV-2 Strains at 28 dpi

The systemic response of CD4^+^ T cells to homologous and heterologous PRRSV-2 strains was analyzed at 28 dpi. The results are depicted in Figure 3 and they include the following parameters: Proliferation and differentiation of Th and Treg cells (Figure 3A,B); IFN-γ, TNF-α, and IL-2 production of CD4^+^ cells; also including differentiation from naïve cells (CD8α^−^CCR7^+^) into T_CM_ (CD8α^+^CCR7^+^) cells and T_EM_ (CD8α^+^CCR7^−^) cells (Figure 3C–E). CC-chemokine receptor 7 (CCR7) expression indicates whether the cells are homing to the tissue (CCR7^−^) or homing to the lymph nodes (CCR7^+^). While the lack of an anti-porcine CD45RO antibody impedes the discrimination of antigen-experienced non-memory and memory T cells, this CD8α/CCR7 differentiation into naïve, T_CM_ and T_EM_ is consistent with previously published research and definitions [14,24,25].

Systemically, the MLV vaccination induced a proliferative response in the Th and Treg cells to the homologous MLV strain and the heterologous HP strain, but these cells did not respond to the LP strain. The infection with LP and HP PRRSV-2 strains induced a broader homo- and heterologous response in the Th cells and Tregs. The homo- and heterologous response to HP in vitro restimulation was generally the highest, but these CD4^+^ T cells from infected pigs also proliferated to heterologous MLV and LP re-stimulation (Figure 3A,B, left graphs). With respect to cytokine production, while CD4^+^ cells did not produce the increased amounts of IL-2 upon homo- or heterologous PRRSV-2 restimulation, they produced TNF-α. However, these CD4^+^ T cells responded most strongly with producing IFN-γ. As seen for the proliferative response of systemic CD4^+^ T cells, MLV induced the strongest heterologous IFN-γ and TNF-α production to HP restimulation. In addition, within the LP and HP animals, a higher frequency of CD4^+^ T cells produced IFN-γ and TNF-α upon homo- and heterologous PRRSV-2 restimulation and their cytokine response seems to be broadly cross-reactive (Figure 3C,D, left graphs). The analysis of the differentiation of CD4^+^ T-cell subsets responding with the proliferation or cytokine production to homo- and heterologous PRRSV-2 strains can be performed in several directions. Therefore, three statistical analyses were performed on these data. First, the response of CD4^+^ T cells were compared to PRRSV-2 restimulation with the response of the unstimulated cells (Media group) WITHIN each in vivo treatment group such as the MLV pigs. This analysis shall answer the question if there is a significant difference between the cells responding to PRRSV-2 and the cells with a background response. In addition, the responding CD4^+^ T cells BETWEEN the in vivo treatment groups with the MOCK pigs were compared. This analysis shall answer the question if the in vivo treatment affected the differentiation of PRRSV-specific CD4^+^ T cells responding to all PRRSV-2 strains (second analysis; Figure 3 right graphs, black letters above bar) or to the homologous PRRSV-2 strain (third analysis; Figure 3 right graphs, colored letters below bar).

The proliferating Th cells from the MOCK pigs did not show any differences in the differentiation between the media and any of the PRRSV-2 strain restimulations. In contrast, the Th cells from the MLV, LP and HP pigs proliferating upon PRRSV-2 restimulation had in nearly all groups significantly fewer naïve cells than when cultured in media (Figure 3A, Naïve). The decreased frequency of naïve cells responding to the PRRSV-2 restimulation is reflected in the increase in responding memory cells. In most cases, the majority of these cells belonged to the central memory compartment (Figure 3A, Central Memory and Effector Memory). These differences within the groups were much less pronounced for Treg cells (Figure 3B, three right graphs). In regard to the frequency of naïve cells within cytokine-producing cells, the cells isolated from the MOCK pigs did not show any differences, but CD4^+^ cells producing IFN-γ or TNF-α belonged, in most cases, to a lower frequency into the naïve cell compartment. While Th cells proliferating upon PRRSV-2 restimulation were mainly central memory cells, IFN-γ and TNF-α producing CD4^+^ cells mainly differentiated into effector memory cells (Figure 3C,D, three right graphs).

A comparison between the in vivo treatment groups also reveals relevant differences. Compared to the MOCK group, both the MLV vaccinated and LP or HP infected groups had significantly less naïve cells within the Th cells proliferating to homologous restimulation (Figure 3A, Naïve colored letters). Comparing their homo- and heterologous response to all PRRSV-2 restimulations, the LP and HP infected groups had even fewer naïve cells compared to the MLV vaccinated group (Figure 3A, Naïve black letters). In contrast to the within group comparison, comparing between the groups shows the most significant difference in the effector memory subsets. For both homologous and heterologous comparison, the MLV and LP infected pigs had significantly more effector memory cells than the MOCK pigs. On top, the HP infected pigs had even more T_EM_ cells than all the other groups. Again, these differences were less pronounced for Tregs (Figure 3B). The between group comparisons for all three cytokines support the results of the differentiation analysis within the proliferating Th cells for the LP and HP infected groups. Compared to the MOCK pigs, cytokine producing CD4^+^ cells from the LP and HP-infected animals had significantly fewer naïve cells (Figure 3C-E, Naïve, black and colored letters). While there were only small changes in the differentiation of IFN-γ and IL-2 producing cells between the MOCK and MLV pigs, TNF-α producing CD4^+^ cells in the MLV pigs had fewer naïve cells than within the MOCK pigs. As for the between group comparisons of the proliferating Th cells, the lower frequencies in the undifferentiated naïve cells are mainly reflected by the increased numbers of effector memory cells. Within IFN-γ and IL-2 producing CD4^+^ cells, the PRRSV-infected LP and HP animals had more effector memory cells and within TNF-α producing CD4^+^ cells, all PRRSV-2 inoculated groups had more T_EM_ cells (Figure 3C–E, three right graphs).

To determine the correlation of the ex-vivo T-cell response with the substantial reduction of viremia between 21 and 35 dpi, T-cell response characteristics were selected with clear significant differences across the virus and inoculation at 28 dpi (Figures 3, 5 and 6) and compared those characteristics to the decrease in viremia. The homologous restimulation data were utilized for all T-cell response characteristics and only included the MLV, LP and HP pigs. Of all the T-cell response characteristics compared, the significant drop in viremia between 21 dpi and 35 dpi only exhibits a significant correlation with Th (Figure 4A) and Treg (Figure 4B) proliferation at 28 dpi.

In conclusion, these data demonstrate that the MLV vaccination induced a CD4^+^ T-cell response to the homologous MLV and the heterologous HP PRRSV-2 strains. Both, the LP and HP PRRSV-2 infection induced broadly reactive CD4^+^ T cells recognizing all three virus strains. The reactive cells displayed enhanced proliferation as well as IFN-γ and TNF-α production. On top, the reactive cells were further differentiated into both memory cell subsets. Using the within group comparison, the proliferating cells were mainly found in the central memory subset. For most other comparisons, the in vivo-induced PRRSV-specific CD4^+^ T cells were mainly effector memory cells. Finally, of all the significantly different quantifiable characteristics of the T-cell response at 28 dpi, only the Th and Treg proliferation exhibit a significant correlation with viremia clearance.

### 3.4. CD3^+^CD8α^+^ T-Cell (CTLs) Immune Response to Homo- and Heterologous PRRSV-2 Strains at 28 dpi

A detailed analysis of the CTL proliferation and cytokine production to PRRSV-2 at 28 dpi is shown in Figure 5. This analysis confirms the data obtained in the time kinetic mentioned above: The CTL response to PRRSV-2 was lower than the response of CD4^+^ cells. While the trend of CTL proliferation and cytokine production was similar to CD4^+^ cells, the significant differences between the MOCK and PRRSV-inoculated animals could only be found for proliferating CTLs from the LP pigs and for IFN-γ producing CTLs mainly upon HP re-stimulation (Figure 5A–D, left graphs). The differentiation analysis of PRRSV-specific CTLs using the between group comparison revealed that in vivo inoculation significantly increased the presence of homo- and heterologous tissue-homing CCR7^−^ CTLs (Figure 5, right graphs): First, the proliferating CTLs from all the PRRSV-inoculated groups had a higher frequency of CCR7^−^ cells compared to the MOCK group. Second, cytokine-producing CTLs of the LP and HP infected pigs had more CCR7^−^ cells compared to the MOCK and MLV pigs.

These results demonstrate that PRRSV-2 vaccination or infection does not induce IL-2 and TNF-α production in CTLs and that these CTLs also show a weak proliferative and IFN-γ response. The most remarkable in vivo effect of PRRSV-2 on CTLs is the induction of tissue-draining CCR7^−^ CTLs. Within the proliferating cells, the MLV vaccination and both PRRSV-2 infections induced CCR7^−^ CTLs. Within cytokine-producing cells, only the LP and HP PRRSV-2 infections induced CCR7^−^ CTLs.

### 3.5. TCR-γδ T Cells Ex Vivo Immune Response to Heterologous PRRSV-2 Strains at 28 dpi

Compared to TCR-αβ T cells, the systemic TCR-γδ T cells show a less significant proliferative and cytokine response to PRRSV at 28 dpi (Figure 6).

Compared to unstimulated media cells, restimulation with the HP strain resulted in significantly more proliferation and IFN-γ production of TCR-γδ T cells than in all the inoculated groups (Figure 6A,B, left graphs). As for CTLs, TCR-γδ T cells from the MLV pigs tend to proliferate and produce IFN-γ to the homologous MLV and heterologous HP restimulation. In contrast, TCR-γδ T cells from the LP and HP infected pigs show a broader heterologous response to all PRRSV-2 strains (Figure 6A,B, left graphs). The amount of systemic IL-2 and TNF-α production by TCR-γδ T cells at that time point did not show any significant differences between the MOCK and PRRSV-inoculated groups (Figure 6C,D, left graphs).

The most significant changes were once more found in the differentiation of TCR-γδ T cells, especially when studying the proliferating TCR-γδ T cells. The data on the differentiation of porcine TCR-γδ T cells are scarce. To the author’s knowledge, this study represents the first analysis of the differentiation of porcine TCR-γδ T cells using CD8α in combination with the lymph node homing chemokine receptor CCR7. While recent manuscripts show a more flexible profile for certain subsets of TCR-γδ T cells [24,26,27,28], CD8α has been described to distinguish naïve (CD8α^−^) and antigen-experienced/ memory (CD8α^+^) TCR-γδ T cells [18,28]. Our data support this role for CD8α in two ways: First, in some cases, the MLV vaccination, but mainly the LP and HP PRRSV-2 infection, increased the frequency of CD8α^+^ cells within proliferating TCR-γδ T cells (Figure 6A, CCR7^+^CD8α^+^ and CCR7^−^CD8α^+^, black and colored letters); and second, within each group and compared to the unstimulated media cells, CD8α^−^ TCR-γδ T cells responded generally less to PRRSV-2 but CD8α^+^ TCR-γδ T cells responded more to homo- and heterologous PRRSV-2 restimulation (Figure 6A, four right graphs). The increased frequency of CD8α^+^ TCR-γδ T cells was visible within both the CCR7 subsets (Figure 6A, CCR7^+^CD8α^+^ and CCR7^−^CD8α^+^).

The overall expression of CCR7 and CD8α within the cytokine producing TCR-γδ T cells revealed additional interesting differences. While TNF-α and IL-2 producing cells seem to be relatively evenly spread over the four CD8α/CCR7 subsets, IFN-γ producing TCR-γδ T cells show a clear pattern. While both the CCR7^+^ subsets have only very few IFN-γ producing TCR-γδ T cells, CCR7^−^CD8α^+^ TCR-γδ T cells have with ~20% the second highest frequency in IFN-γ producing cells. However, the majority of IFN-γ producing TCR-γδ T cells can be found within the CCR7^−^CD8α^−^ subset (Figure 6B, four right graphs). PRRSV seems to have a rather minor effect on the differentiation of these IFN-γ-producing TCR-γδ T cells (Figure 6B, four right graphs). In contrast, TNF-α and IL-2 producing TCR-γδ T cells, especially upon HP restimulation, seem to shift from CCR7^−^CD8α^−^ to CCR7^+^CD8α^+^ TCR-γδ T cells (Figure 6C,D, four right graphs).

While CD4^+^ cells from the MOCK pigs have a high frequency of naïve (= CCR7^+^CD8α^−^) PRRSV-responsive cells (Figure 3A–E, MOCK, Naïve), TCR-γδ T cells responding to PRRSV nearly lack this cell type (Figure 6A–D, CCR7^+^CD8α^−^). In contrast, these TCR-γδ T cells have up to 80% CCR7^−^CD8α^−^ cells (Figure 6, right column). This subset is nearly non-existent in CD4^+^ T cells and was therefore excluded in the analysis. The discrepancy in the CCR7 expression within the naïve CD8α^−^ population between the CD4^+^ and TCR-γδ T cells led the authors to further decipher the effect of PRRSV on the CCR7 expression within TCR-γδ T cells as shown in Figure 7. Figure 7A compares the CCR7 expression of non-proliferating (dashed line) and proliferating (solid line, filled histogram) TCR-γδ T cells from a high responder animal within each inoculation group. Figure 7B,C show the results for all animals in these groups regarding the mean fluorescence intensity (MFI) and percent CCR7^+^, respectively. Our results demonstrate that within the MOCK and all PRRSV-inoculated groups, and upon homologous PRRSV-2 restimulation, the proliferating TCR-γδ T cells have a significantly increased CCR7 MFI as well as a higher frequency of CCR7^+^ cells. This analysis shows that irrespective of a previous contact to the pathogen, TCR-γδ T cells proliferating to PRRSV-2 upregulate the expression of CCR7.

These data demonstrate that TCR-γδ T cells contribute to the heterologous response to PRRSV-2 strains. PRRSV-specific TCR-γδ T cells mainly react with proliferation and a limited production of IFN-γ. Of note, and in contrast to TCR-αβ T cells, PRRSV-2 activated TCR-γδ T cells upregulate the expression of the lymph node homing chemokine receptor CCR7.

### 3.6. The T-cell Response to PRRSV-2 in the Tracheobronchial Lymph Node at 63 Days Post Infection

In addition to the systemic T-cell response, the proliferative response of the T cell subsets isolated from the tracheobronchial lymph nodes was analyzed at necropsy (= 63 dpi, Figure 8. T cells from the MOCK pigs did not respond to any in vitro restimulation. In contrast to the systemic T cell response within the MLV pigs, the lymph node T cells at 63 dpi did not respond to HP restimulation. However, these regional T cells from the MLV pigs showed only a response to homologous PRRSV-2 restimulation. The regional T cells from the LP and HP pigs displayed a non-significant response to the MLV strain; but they responded with a strong proliferative response to both, homo- and heterologous LP and HP PRRSV-2 strains (Figure 8A). The regional Th response showed a similar picture with a non-significant homologous response of Th cells from the MLV pigs and a robust Th response in the LP and HP pigs to both homo- and heterologous LP and HP stimulation. As for the systemic Th response at 28 dpi, the HP Th response was stronger than the LP proliferative response (Figure 8B). While also Tregs from the MOCK pigs responded to the LP and HP restimulation, the most significant response to the LP and HP restimulation was seen in regional Tregs from the LP and HP pigs (Figure 8C). Consistent with peripheral blood, the CTL response to PRRSV-2 restimulation was weaker compared to Th cells. In addition, only the LP pigs displayed a significant CTL proliferative response and they also displayed cross-reactivity with the HP PRRSV-2 strain (Figure 8D). TCR-γδ T cells from the LP and HP pigs exhibited a significant increase in proliferation compared with the media and exhibited significant cross-reactivity between the two LP and HP strains (Figure 8E).

These results demonstrate that while the systemic T-cell response in blood has declined to a low level by 56 dpi, the T-cell response in the tracheobronchial lymph node is still ongoing. While the heterologous response of T cells from the MLV pigs to the HP PRRSV-2 strain could not be found in the tracheobronchial lymph node, some of the most important results obtained in blood at 28 dpi are consistent with the regional T-cell response in this lung draining lymph node: i) As for the systemic response, the regional T-cell response in the lung draining lymph nodes is dominated by Th cells; and ii) in vivo infection with the LP and HP PRRSV-2 strains induced the strongest T-cell response; and these responding T cells have a robust heterologous response to the other LP or HP PRRSV-2 strain. Regarding TCR-γδ T cells, while the frequency of TCR-γδ T cells is relatively low in the tracheobronchial lymph node (generally <10% of all T cells, Figure 8 and Figure 9), these cells showed the highest frequency of responding cells within their subset: 40–60% of TCR-γδ cells from the LP and HP pigs proliferated upon LP and HP restimulation.

### 3.7. Distribution of Immune Cell Subsets and Prevalence of PRRSV-2 in the Lung and in Lymphoid Tissues at 63 dpi

At nine weeks post infection, the pigs were sacrificed. Then, the lung and lymphoid tissues were collected to analyze the distribution of the immune cells and the prevalence of PRRSV-2 (Figure 9). While viremia was absent in the LP and HP pigs at 63 dpi, PRRSV-2 was still present in macrophages in the peripheral tissues from the pigs of all three PRRSV-2 inoculated groups—MLV, LP and HP. In addition, the MLV animals displayed PRRSV-2^+^ dendritic cells and moDCs in the tonsils and lung. Using flow cytometry, the lymph nodes and BAL were the only compartments, in which PRRSV-2 could not be detected (Figure 9A).

The distribution of the immune cell populations in the lung and lymphoid tissues was analyzed utilizing a novel 9-color FCM immunophenotyping panel (Figure 9A–E, only myeloid and T-cell data shown). The frequencies of the T and myeloid cells were relatively consistent between the in vivo inoculation groups (Figure 9B). Independent of the PRRSV-inoculation groups, it is notable that not only in the lymphoid tissues, but also within the lung, a high frequency of ~20–40% of leukocytes represent T cells (Figure 9B).

Figure 9C–E study the distribution of CD4^+^, CTL and TCR-γδ cells within the analyzed tissues. For CD4^+^ T cells, the inoculated pigs display slight differences from the MOCK in the mediastinal lymph nodes with the MLV pigs having more CD4^+^ T cells, HP pigs having higher T_CM_, and LP pigs having more Tregs (Figure 9C). As expected, the naïve and central memory CD4^+^ cells mainly populated the lymphoid tissues. In contrast, the effector memory cells dominated the lung and BAL. While these CD4^+^ T cells were rather consistent between the infection groups, PRRSV-2 vaccination or infection clearly resulted in a shift in the composition of T cells. Within the T cells, the frequency of CTLs increased in the mediastinal lymph node, lung and BAL (Figure 9D). In contrast, the frequency of TCR-γδ T cells in these tissues decreased in PRRSV-inoculated animals (Figure 9E). Of note, the MLV pigs had a higher percentage of CD4^−^CD8α^+^FoxP3^+^ Tregs in the tonsils, mediastinal and tracheobronchial lymph nodes. Regarding TCR-γδ T cell differentiation, the HP pigs experienced a shift in the cells from CD8α^−^CD27^−^ into the CD8α^+^ compartments in the lung or mediastinal lymph nodes (Figure 9D–E).

In conclusion, PRRSV-2 has been cleared from the lung in the LP and HP infection which is accordance with the absence of viremia in those pigs. In the MLV vaccinated pigs on the other hand, PRRSV-2 was still present in the blood and it could be also detected in their lungs as well as in the spleens and tonsils. Regarding the effect of the PRRSV-2 vaccination or infection on the distribution of T cells subsets in the lung and lymphoid tissues at 9 weeks post infection, the most remarkable finding was that even at such a late time point, in the MLV, LP and HP pigs, more CTLs and less TCR-γδ T cells were present in the lung-draining mediastinal lymph node, the lung itself, and the BAL.

## 4. Discussion

The overall goal of this study was to provide a detailed insight into the T-cell response to homo- and heterologous PRRSV-2 strains. To provide this information, this study followed the infection, clinical pathology and T-cell immune response of PRRSV-vaccinated or infected animals over nine weeks.

PRRSV-2 infection leads to viremia within one week. Viremia is usually cleared by week 5–7 with some rebounds at later time points [2,29]. In lymphoid tissue on the other hand, PRRSV-2 can persist over several months [2,29]. The PRRSV-2 inoculations led to a similar viremia timeline and persistence of PRRSV-2 in the lymphoid tissue: After clearing viremia in the LP animals at 49 dpi and in the HP animals at 35 dpi, PRRSV-2 rebounded in the blood until 56 dpi. After that, PRRSV-2 could still be detected at 63 dpi in their spleens (Figure 9A). At this time point, the MLV pigs were still viremic and besides lymphoid tissues, PRRSV-2 was still present in the lung. Clinical pathology was absent in the MOCK and MLV pigs. In contrast, the LP and HP pigs had some pathology including fever, lethargy and reduced body weights. These clinical observations were rather mild. The limited PRRSV pathology can be explained by the clean study conditions limiting the risk of secondary infections. Despite some studies which report pig deaths in PRRSV infection studies [30,31], the mild pathology is also consistent with several PRRSV studies performed under ABSL-2 conditions [32,33,34,35]. Overall, the PRRSV-2 infection and clinical pathology data are consistent with the PRRSV vaccination and infection trials described in the literature. This consistency provides the relevant basis for the interpretation of the focus of this study—The T-cell immune response to PRRSV-2.

To address the substantial knowledge gap in the T-cell response to PRRSV-2, the presented in vivo trial with extensive in vitro PRRSV-2 restimulation assays were combined. In contrast to the ex vivo analysis of immune cells, this kind of restimulation allows the analysis of PRRSV-specific T-cells. To provide further detail, this study used up to nine-color polychromatic flow cytometry to provide an in-depth analysis of this T-cell response to PRRSV-2.

All analyzed T-cell subsets seem to be involved in both the homologous and heterologous immune response to PRRSV-2 (Figure 3, Figure 5 and Figure 6). While the MLV PRRSV-2 strain belongs to the PRRSV-2 lineage 5.1, the LP and HP strains are PRRSV-2 lineage 1 strains. Our data demonstrate that genomic analyses, especially when only based on a partial genome such as the PRRSV ORF-5 sequence, are not suitable to reflect the complexity of the heterologous immune response to PRRSV-2. The complexity of this heterologous response is best explained using the heterologous Th response. In blood, the HP pig Th cells display a strong proliferative response to MLV restimulation and the MLV pig Th cells exhibit a significant proliferative response to the HP strain. Additionally, the isolated PBMCs in HP pigs do not exhibit a significant Th proliferative response to the LP strain. However, the tracheobronchial lymph node HP pig Th proliferative response to the LP strain is significant. The question is: What accounts for the changes in cross-reactivity in Th cells between 28 dpi and long-term T_CM_ cells? Our hypothesis to explain these changes in cross-reactivity is grounded in the basic immunology concept of affinity maturation [36]. This concept states that T cells with a higher affinity for the antigen will proliferate and differentiate and the cells with a weaker interaction will die. In this case, the long-term memory population of the HP pigs in the tracheobronchial lymph nodes specificity for the HP strain is greater than during the peak of infection.; The greater specificity for this homologous strain reduces the cross-reactivity with MLV. The same is true for the MLV pigs cross-reactivity with the HP strain. The MLV pigs’ long-term memory population does not exhibit the cross-reactivity with the HP strain evidenced at 28 dpi. This hypothesis would indicate that affinity maturation might represent a challenge for long-term vaccine efficacy to heterologous PRRSV-2 strains. It also indicates that, since this mechanism is host-based, it cannot be predicted by using a PRRSV-2 sequencing tool. A detailed analysis of the immune response can provide a much better prediction of this anti-PRRSV-2 response as shown below.

The overall pattern of lymphocyte proliferation upon PRRSV-2 infection was first determined over 20 years ago. The PRRSV-specific lymphocyte proliferation was detectable starting at 4 weeks post-infection. This response had two peaks: One at four and one at seven weeks post-infection. Afterwards, it declined and was absent at 11 weeks post infection. In addition, Bautista et al. determined that CD4^+^ T cells exhibited a stronger proliferative response than CD8^+^ T cells [37]. Recent papers have determined phenotypic changes of T cells within PBMCs upon PRRSV-2 infection. However, those studies do not address an antigen-specific proliferative response [25,38,39]. Our own data confirm these two proliferation peaks and the strong CD4^+^ response. This study also provides additional insight into the potential cause of the two proliferation peaks. First, as shown in Figure 2, while T-cell proliferation was detectable at 14 dpi, it mainly peaked at 28 dpi. Second, CD4^+^ Th cells were the main responders at this first 28 dpi peak. Third, HP PRRSV-2 infection also led to a second peak in T-cell proliferation at 56 dpi. This second peak can be best explained by the high proliferation of TCR-γδ T cells (Figure 2F). These cells dominated the T-cell response in all PRRSV-2 inoculation groups at the later time points. At 14 and 28 dpi, TCR-αβ T cells were the main responders but 42 and 56 dpi, TCR-γδ T cells exhibited the strongest proliferative response.

While TCR-γδ T cells are postulated to produce IFN-γ at early time points ([2], Figure 6), the limited research in pigs on the specific role of TCR-γδ T cells in combating PRRSV-2 leaves one important question open: What role do these TCR-γδ T cells play at such a late time point? Our data support that these TCR-γδ T cells play an important role in the immune response to PRRSV-2 in lymphoid tissue. First, the systemic TCR-γδ T cells are most active when viremia is mainly cleared, but at these late time points PRRSV-2 is still persisting in lymphoid organs, such as the spleen or tonsils (Figure 2F and Figure 9A). Second, in all groups but especially in the LP group, TCR-γδ T cells responding to PRRSV-2 upregulate CCR7 (Figure 7): This chemokine receptor is responsible to direct these TCR-γδ T cells into lymphoid tissues. Third, while the overall TCR-γδ T cells frequency decreased in the mediastinal lymph nodes of PRRSV-inoculated animals, the response of the TCR-γδ that were present in the lymph nodes was tremendous. Upon PRRSV-2 restimulation, TCR-γδ T cells from the LP and HP-infected pigs responded with ~40–60% proliferation. In comparison, Th cells responded to ~5–20%. While the frequent Th certainly play a role in the late immune response to PRRSV-2 as well, the three points mentioned above provide a strong indication that TCR-γδ T cells are highly involved in the immune response to PRRSV-2 within the lymphoid tissues. The role TCR-γδ T cells play in this response is still unclear. As their TCR-αβ counterparts, TCR-γδ T cells can have multiple functions. These functions include immunosuppressive and immunostimulatory capacities [3]. Therefore, it still needs to be determined if TCR-γδ are either involved in downregulating the immune response to PRRSV-2 which might promote PRRSV-2 persistence in lymphoid tissue, or those TCR-γδ are helping in the clearance of PRRSV-2 from those tissues. In any case, TCR-γδ T cells seem to play a critical role in PRRSV-2 persistence in lymphoid tissues.

These studies on the differentiation and homing receptors CD8α and CCR7 revealed an interesting peculiarity of TCR-γδ T cells: As their CD4^+^ T-cell counterparts, TCR-γδ T cells seem to upregulate CD8α upon antigen-exposure (Figure 6, especially A). In contrast, while naïve and central memory CD4^+^ T cells are CCR7^+^ and downregulate the CCR7 expression upon further differentiation into the effector memory cells [14], the naïve TCR-γδ T cells seem to be CD8α^−^CCR7^−^ (Figure 7, non-proliferating cells; and Supplementary Figure 4, bottom left plot). This means, the naïve TCR-γδ T cells migrate between the blood and tissue. This phenomenon would attribute a potentially critical role to TCR-γδ T cells in the early response to PRRSV at the site of infection. Upon stimulation, these naïve CD8α^−^CCR7^−^ TCR-γδ T cells seem to acquire different homing patterns probably depending on their role in the immune response to PRRSV: On the one side, IFN-γ-producing TCR-γδ T cells stay CCR7^−^. On the other side, TNF-α-producing and proliferating TCR-γδ T cells upregulate their CCR7 expression (Figure 6A,C, CCR7^+^CD8α^+^; and Figure 7, proliferating cells). These CCR7^+^ TCR-γδ T cell subset can thereby acquire the ability to drain into the lymphatic system during the activation/ differentiation process. Combined with the strong proliferative response of TCR-γδ T cells in blood, this lymph node homing ability indicates an important role for TCR-γδ T cells in the immune response to PRRSV in the lymphatic tissue during persistent PRRSV infection. The details of the role of TCR-γδ T cells in PRRSV are currently unknown. IFN-γ-producing TCR-γδ T cells in the lung probably contribute to the anti-viral response by macrophage activation. It is speculated that lymph node draining TCR-γδ T cells could promote inflammation by their TNF-α production and they could additionally be involved in TCR-αβ T-cell stimulation by potential antigen presentation.

The differentiation of porcine TCR-γδ T cells has so far only been performed using CD8α in combination with CD27 [24]. Talker et al. [24] showed that TCR-γδ T cells in young animals are CD8α^−^. This result, combined with further phenotypic and functional studies of TCR-γδ T cells [26], confirm the data obtained in this study. They indicate that naïve TCR-γδ T cells are CD8α^−^. Within these most probably naïve CD8α^−^ TCR-γδ T cells, the vast majority expressed CD27. “Notably, a CD27^−^ … phenotype could only be detected in combination with CD8α expression.” In contrast, antigen-experienced/ memory TCR-γδ T cells included CD27^−^ cells [24]. This indicates that while CCR7 and CD27 are co-expressed in CD4^+^ T cells [25,40], these markers are inversely expressed in porcine TCR-γδ T cells. The naïve TCR-γδ T cells are CD27^+^ and CCR7^−^ and they upregulate CCR7 but downregulate CD27 during their activation/ differentiation process. Studies in humans show a similar CD27 expression pattern in TCR-γδ T cells from peripheral blood. The naïve and central memory TCR-γδ T cells are CD27^+^ and effector memory TCR-γδ T cells are CD27^−^ [41,42,43]. Berglund et al. studied the differentiation of TCR-γδ T cells from cord blood using both markers, CD27 and CCR7. Their results confirmed our observations in swine. While the majority of naïve (CD45RO^−^) TCR-γδ T cells within human cord blood express CD27, they are CCR7^−^ [40]. This is the first study showing that in contrast to CD27, porcine naïve TCR-γδ T cells are CCR7^−^ and they can upregulate CCR7 during their activation/ differentiation process.

While TCR-γδ T cells dominate the late systemic response to PRRSV-2, TCR-αβ T cells dominate the early immune response during viremia. These TCR-αβ T cells mainly consist of CD4^+^ Th and Treg cells and CTLs. CD4^+^ T cells responding to PRRSV-2 represent up to 10% of all T cells. In contrast, proliferating CTLs upon PRRSV-2 restimulation are less than 1% of all T cells. PRRSV-specific CTLs also produce less cytokines. Only IFN-γ production was significantly increased upon restimulation with HP strains. While the cytotoxic activity of CTLs was not examined here, Costers et al. demonstrated that CD3^+^CD8α^hi^ proliferate as early as 14 dpi, but those cells did not exhibit cytotoxic activity until 49 dpi [44]. Additionally, Lamontagne et al. demonstrated increased CD8α^high^ subsets in the spleen and blood after PRRSV-2 infection until 60 dpi [45]. The analysis of the lymph node and tissue homing of CTLs shed light on a specific role of CTLs in combating PRRSV-2. In contrast to TCR-γδ T cells, PRRSV-specific CTLs from in the MLV, LP and HP animals significantly increased the frequency of tissue-homing CCR7^−^ CTLs (Figure 5). This increased tissue-homing of systemic CTLs is substantiated by the increased frequency of CTLs in the lungs and BAL of PRRSV-2 vaccinated or infected animals (Figure 9D). These data indicate that while the systemic response of CTLs might be secondary to CD4^+^ cells, CTLs might play a crucial role in combating PRRSV-2 at the site of infection, the lung and BAL.

In addition to the potentially vital roles of TCR-γδ T cells in the lymph nodes and CTLs in the lung and BAL, CD4^+^ Th and Treg cells are important to the immune response against PRRSV-2.

The balance between these two important CD4^+^ T-cell subsets is crucial for an effective and healthy immune response. The two research groups reported that PRRSV-2 stimulates Tregs [8,10,11,12,46]. Based on those reports, it has been speculated that the PRRSV-2 is shifting that balance of the CD4 T-cell response towards Tregs. In contrast, a third research group did not find an active role of Tregs in PRRSV-induced immunosuppression [47]. Tregs have the ability to suppress various parts of the immune response [4,6] and this immunosuppression can then lead to prolonged pathogen persistence [48]. While our study confirmed that PRRSV-2 is stimulating Tregs (Figure 2C and Figure 3B), the important question is if this stimulation is strong enough so that PRRSV-2 can shift the Th/Treg balance into an immunosuppressive state. To further investigate this balance, the authors studied the ratio of Th and Treg cells responding to PRRSV-2 over time. Our data show that prior to PRRSV-2 infection (= 0 dpi), or in the absence of PRRSV-2 (= MOCK animals), the ratio of Th/Treg cells responding to PRRSV-2 is on average roughly between 0.5 and 2. Upon PRRSV-2 inoculation, it took until 28 dpi to shift this ratio above the two thresholds. This means that it took 28 days to shift the CD4 immune response into an inflammatory state. On the other hand, PRRSV-2 inoculation was at no time able to shift Th/Treg ratio below the normal 0.5–2 ratio. The delayed start of the Th/Treg ratio confirms an immunosuppressive capacity of PRRSV-2. While Treg activation by PRRSV-2 supports an active role of this subset in PRRSV-2, the lack of induction of a decreased Th/Treg ratio rather challenges the speculation that Tregs are responsible for PRRSV-2 immunosuppression. Therefore, this study concluded that, unfortunately, the data were not able to end the discussion if Tregs play an active or passive role in PRRSV-2 immunosuppression.

Regarding the Th response, as mentioned above, the systemic proliferative response of Th helper cells was the strongest of all analyzed T-cell subsets at 14 and 28 dpi (Figure 2). At 28 dpi, the systemic CD4^+^ cells had also the strongest induction of IFN-γ and TNF-α production (Figure 3C,D), consistent with previous PRRSV-2 research for IFN-γ production in PBMCs using ELISPOT or intracellular cytokine staining [33,49,50,51,52,53,54,55,56]. This strong Th response is peaking at the time at which PRRSV-2 viremia is dropping substantially for all PRRSV-inoculated groups. This timing indicated a possible role for Th cells in clearing PRRSV-2 viremia. To further investigate this indication, this study performed a correlation analyses between the CD4 T-cell response and the viremia drop before and after 28 dpi—Δviremia (21–35 dpi). These analyses substantiated the central role of CD4 T-cells in the clearance of PRRSV-2 viremia: The Treg and Th responses were the only two immune parameters at 28 dpi that significantly correlated with the drop in PRRSV-2 viremia (Figure 4). While Tregs are immunosuppressive, Th cells are vital players in combating viruses. Especially, IFN-γ and TNF-α production by Th cells are important immune mechanisms in anti-viral responses. Therefore, the data strongly support that Th cells play a central role in clearing PRRSV-2 viremia.

The Th cells play a critical role in directing the humoral immune response. In that respect, the MLV, LP and HP inoculated pigs all produced anti-PRRSV-2 systemic IgG levels starting at 14–21 dpi. While the production of neutralizing antibodies was delayed for the MLV strain, the LP and HP infected pigs developed neutralizing antibodies within 7 and 14 dpi, respectively. An in-depth analysis of this early humoral immune response is planned to validate our findings. These data will then be addressed in another manuscript.

Studying the differentiation of Th cells responding to PRRSV-2, it seems that Th cells play additional roles in both the lymphoid and lung tissues. On the one side, Th proliferating upon PRRSV-2 restimulation mainly consist of lymph node homing central memory cells. On the other side, the PRRSV-2 infection mainly leads to the induction of cytokine-producing effector memory Th cells which are migrating into the tissue. The importance of Th cells for responding to PRRSV-2 infection in the lymphoid tissue is additionally supported by the high frequency of PRRSV-specific Th cells within the draining mediastinal lymph node (Figure 8B). For the lung and BAL, this study did not perform restimulation assays. Therefore, PRRSV-specific Th in those tissues could not be determined. The overall frequency of Th in the lung and BAL did not increase in PRRSV-inoculated animals. While this might seem to contradict a role of Th in those tissues, the PRRSV-specific effector memory Th cells mainly responded by cytokine production, not proliferation. This lack in proliferation can explain the lack in an increased frequency of Th cells within the lung and BAL. Nevertheless, the systemic data on the induction of the effector memory Th cells by PRRSV-2 inoculation still provide robust data that Th also play an important role in the anti-PRRSV-2 response in the lung and BAL. In summary, our data show that Th cells can play a vital role in the anti-PRRSV-2 response in all relevant parts of the body—the blood, lymphoid organs, the lung and BAL.

## 5. Conclusions

Our data provide for the first time a complete and in-depth analysis of the T-cell response to PRRSV-2 (Figure 10). The obtained data demonstrate an important role for T cells in the homo- and heterologous immune response to PRRSV-2: TCR-γδ T cells seem to be mainly involved in the immune response within the lymphoid tissues; CTLs probably play a crucial role in the lung and BAL; and Th cells seem to have a central role in combating PRRSV-2 in blood as well as in the lymphoid and lung tissues. With the Th response at 28 dpi, a potential immune correlate of protection has been identified. These immune correlates of protection can be used as a tool to predict the vaccine-efficacy against a pathogen, such as PRRSV-2. Thereby, studying the detailed immune response to homo- and heterologous PRRSV-2 strains provides essential information to improve the assessment of PRRSV-2 vaccines. A better understanding of the immune response to PRRSV-2 will additionally provide essential information to facilitate the development of PRRSV-2 vaccines providing protection against the various heterologous PRRSV-2 strains.

## Figures and Tables

**Figure 1 viruses-11-00796-f001:**
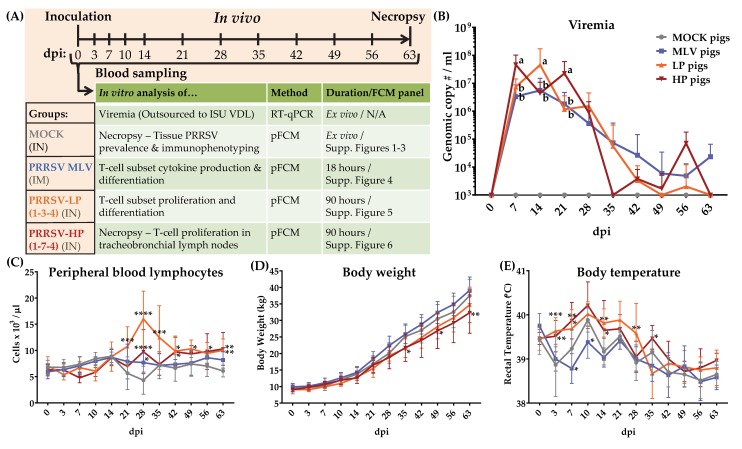
PRRSV-2 inoculation resulted in PRRSV-2 pathology and/or viremia. (**A**) Study design (as described in Materials and Methods); (**B**) viremia for weekly serum samples (qPCR performed by ISU VDL) comparing differences between PRRSV-2 inoculated animals with a 2-way ANOVA and Fisher’s LSD; groups with dissimilar superscripts (a or b) are significantly different (*p* < 0.05); (**C**) peripheral blood lymphocytes were determined on fresh peripheral blood by a complete blood count; (**D**) the animals were weighed prior to blood sampling at each time point; (**E**) the body temperature was determined at each time point by rectal thermometer prior to blood sampling. Graphs illustrate the means with standard deviation (SD). The data were analyzed using Fisher’s LSD repeated-measures 2-way ANOVA by day compared to the MOCK pigs. **** *p* < 0.0001. *** *p* < 0.001, ** *p* < 0.01, * *p* < 0.05.

**Figure 2 viruses-11-00796-f002:**
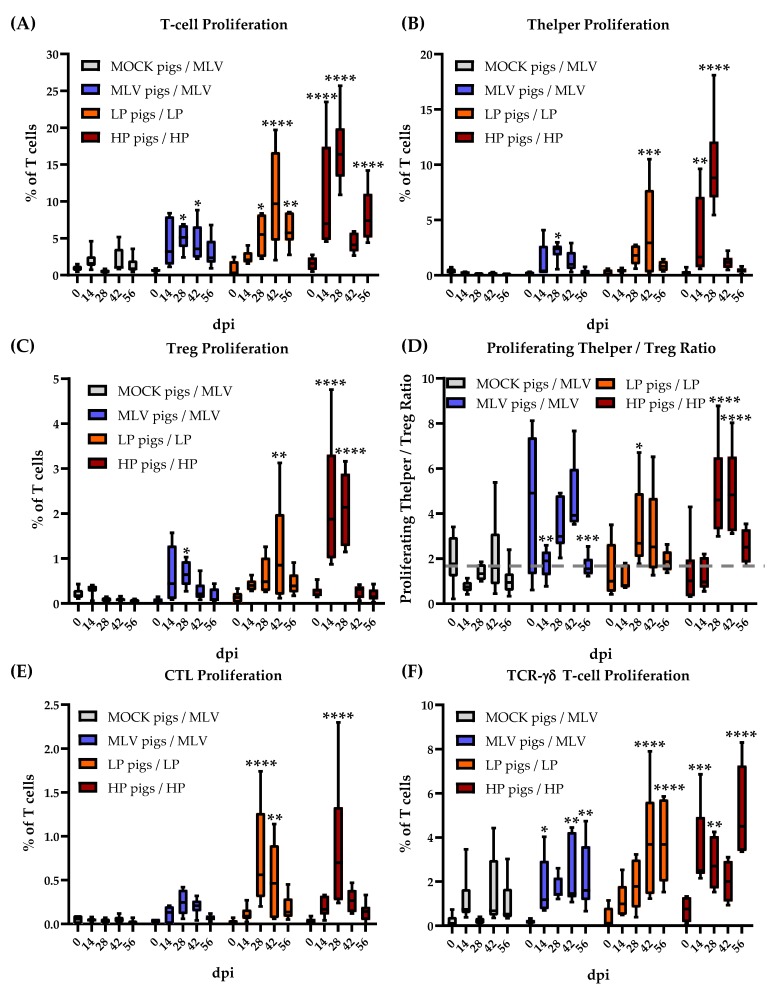
The peripheral blood T-cell immune response peaks at 28 dpi. (**A**) T-cell proliferation; (**B**) Th proliferation; (**C**) Treg proliferation; (**D**) Proliferating Thelper/Treg ratio with the grey dashed line representing the mean (value = 1.79) of inoculated at Day 0 and MOCK at all time points (**E**) CTL proliferation; (**F**) TCR-γδ T-cell proliferation with homologous viral restimulation for 90 h. Proliferation was quantified as the percentage proliferating of all T cells. The data were analyzed using Fisher’s LSD repeated-measures 2-way ANOVA comparing all time points within an in vivo inoculation to 0 dpi. *n* = 6 for all groups except MOCK pigs/MLV, 28 dpi (*n* = 4). **** *p* < 0.0001. *** *p* < 0.001, ** *p* < 0.01, * *p* < 0.05.

**Figure 3 viruses-11-00796-f003:**
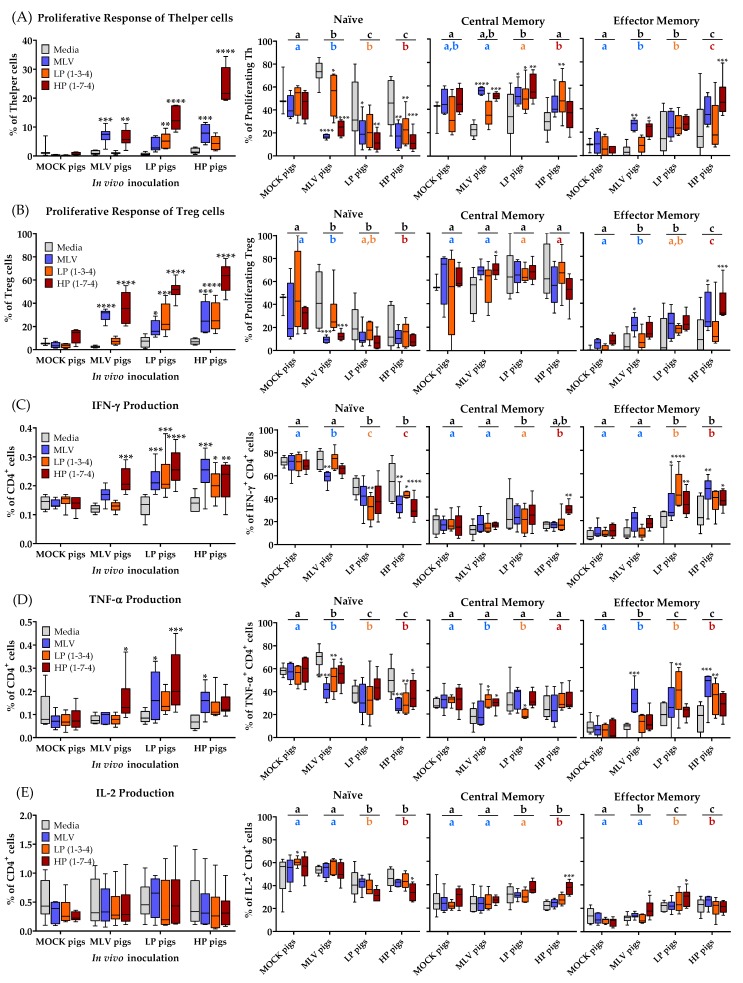
PRRSV-2 inoculation increases Th and Treg proliferation and IFN-γ production. (**A**) Proliferative response of Th cells and the differentiation of proliferating Th cells; (**B**) Proliferative response of Treg cells and the differentiation of proliferating Treg cells; (**C**) IFN-γ production and the differentiation of IFN-γ^+^ CD4^+^ T cells; (**D**) TNF-α production and the differentiation of TNF-α^+^ CD4^+^ T cells; (E) IL-2 production and the differentiation of IL-2^+^ CD4^+^ T cells. Proliferation was quantified as the percentage proliferating of all Th or Treg cells. The data were analyzed using Fisher’s LSD repeated-measures 2-way ANOVA comparing all the viral restimulations within an in vivo inoculation to media. *n* = 6 for all groups except in (**A**) and (**B**) where the MOCK pigs with media (*n* = 3), and the MOCK pigs with MLV/LP/HP (*n* = 4). **** *p* < 0.0001. *** *p* < 0.001, ** *p* < 0.01, * *p* < 0.05. The letters above the graphs compare the different treatment groups. The black letters compare the means of all three viral restimulation. The colored letters compare the means for homologous restimulation (MOCK pigs/MLV restimulation). The groups with dissimilar letters are significantly different (*p* < 0.05).

**Figure 4 viruses-11-00796-f004:**
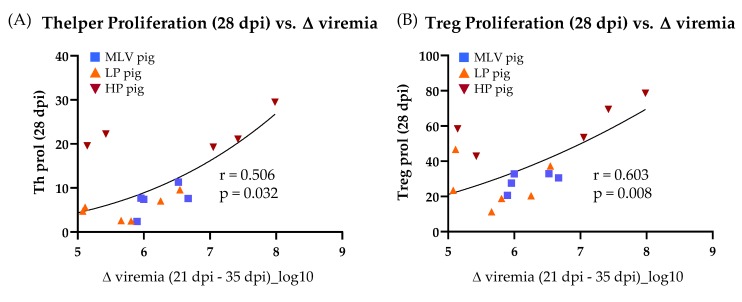
Viremia reduction correlates with Th (**A**) and Treg (**B**) proliferation at 28 dpi. The values for individual pigs are plotted comparing proliferation to the change in viremia from 21 dpi to 35 dpi. Two animals (one MLV and one HP pig) were not plotted because Δ viremia was a negative value or 0. Both were included in the Pearson correlation analysis.

**Figure 5 viruses-11-00796-f005:**
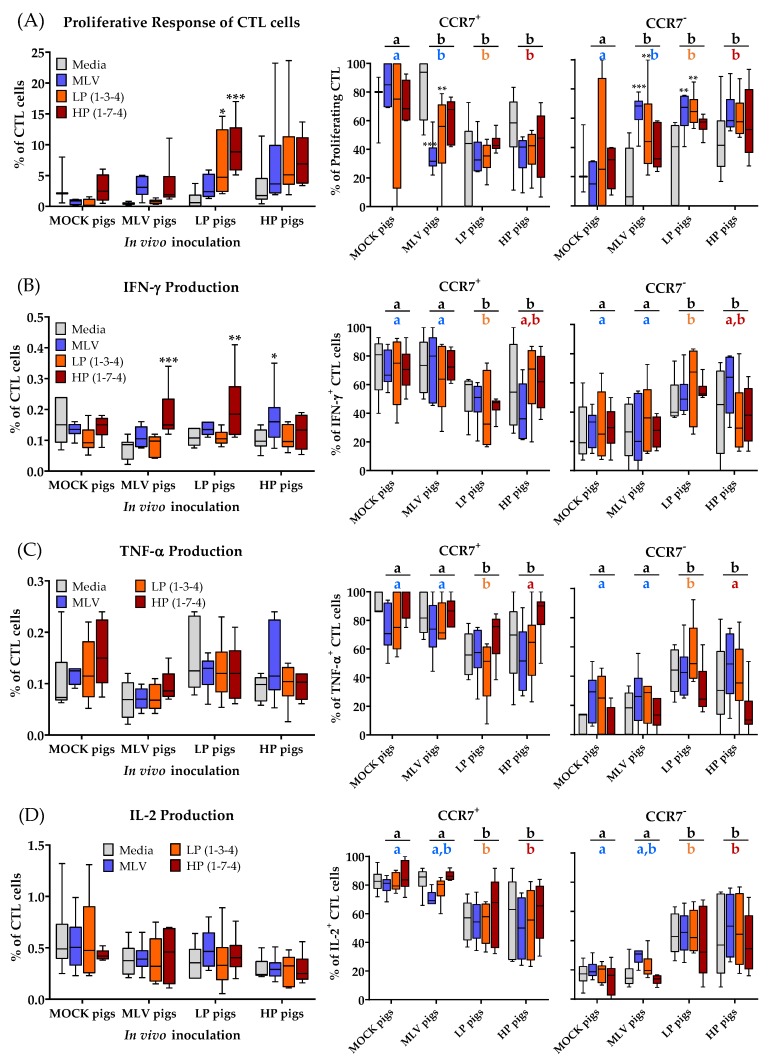
Only low-pathogenic (LP) inoculation induces significant cytotoxic T lymphocyte (CTL) proliferation to viral restimulation. (**A**) Proliferative response of CTLs and the differentiation of proliferating CTLs; (**B**) IFN-γ production and the differentiation of IFN-γ^+^ CTLs; (**C**) TNF-α production and the differentiation of TNF-α^+^ CTLs; (**D**) IL-2 production and the differentiation of IL-2^+^ CTLs. Proliferation was quantified as the percentage proliferating of all CTLs. The data were analyzed using Fisher’s LSD repeated-measures 2-way ANOVA comparing all PRRSV-2 restimulations within an in vivo inoculation to media. *n* = 6 for all groups except in (**A**) where the MOCK pigs with media (*n* = 3) and the MOCK pigs with MLV/LP/HP (*n* = 4). **** *p* < 0.0001, *** *p* < 0.001, ** *p* < 0.01, * *p* < 0.05. The letters above the graphs compare the different treatment groups. The black letters compare the means of all three viral restimulation. The colored letters compare the means for homologous restimulation (MOCK pigs/MLV restimulation). The groups with dissimilar letters are significantly different (*p* < 0.05).

**Figure 6 viruses-11-00796-f006:**
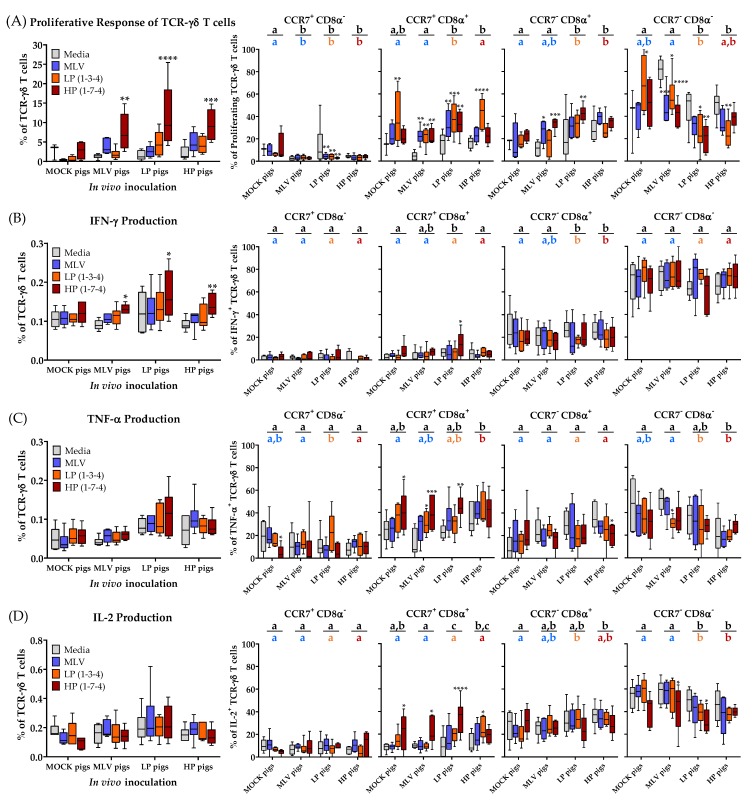
High-pathogenic (HP) virus restimulation results in TCR-γδ T cells proliferation and the differentiation of a CD8α^+^CCR7^+^ subset. (**A**) Proliferative response of TCR-γδ T cells and the differentiation of proliferating TCR-γδ T cells; (**B**) IFN-γ production and the differentiation of IFN-γ^+^ TCR-γδ T cells; (**C**) TNF-α production and the differentiation of TNF-α^+^ TCR-γδ T cells; (**D**) IL-2 production and the differentiation of IL-2^+^ TCR-γδ T cells. Proliferation was quantified as the percentage proliferating of all TCR-γδ T cells. The data were analyzed using Fisher’s LSD repeated-measures 2-way ANOVA comparing all PRRSV-2 restimulations within an in vivo inoculation to media. *n* = 6 for all groups except in (**A**) where the MOCK pigs with media (*n* = 3) and the MOCK pigs with MLV/LP/HP (*n* = 4). **** *p* < 0.0001, *** *p* < 0.001, ** *p* < 0.01, * *p* < 0.05. The letters above graphs compare the different treatment groups. The black letters compare the means of all three viral restimulation. The colored letters compare means for homologous restimulation (MOCK pigs/MLV restimulation). The groups with dissimilar letters are significantly different (*p* < 0.05).

**Figure 7 viruses-11-00796-f007:**
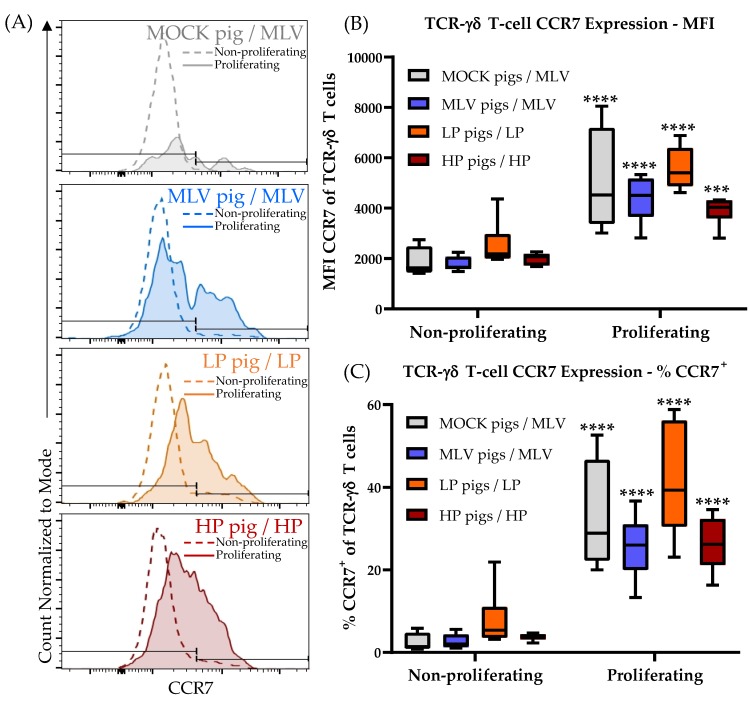
TCR-γδ T cells proliferating to PRRSV-2 upregulate the expression of the lymph node homing chemokine receptor CCR7. (**A**) One high responding pig from each treatment group was selected to demonstrate the differences in CCR7 expression between the proliferating and non-proliferating TCR-γδ T cells with the homologous virus restimulation. The counted cells were normalized to the mode for visual representation; (**B**) the mean fluorescent intensity (MFI) for CCR7 is compared between non-proliferating and proliferating TCR-γδ T cells with homologous virus restimulation; (**C**) The percentage of CCR7^+^ TCR-γδ T cells is compared between non-proliferating and the proliferating TCR-γδ T cells with the homologous virus restimulation; the data were analyzed using Fisher’s LSD repeated-measures 2-way ANOVA comparing non-proliferating to proliferating within an in vivo inoculation. *n* = 6 for all groups except MOCK pigs/MLV (*n* = 4). **** *p* < 0.0001. *** *p* < 0.001, ** *p* < 0.01, * *p* < 0.05.

**Figure 8 viruses-11-00796-f008:**
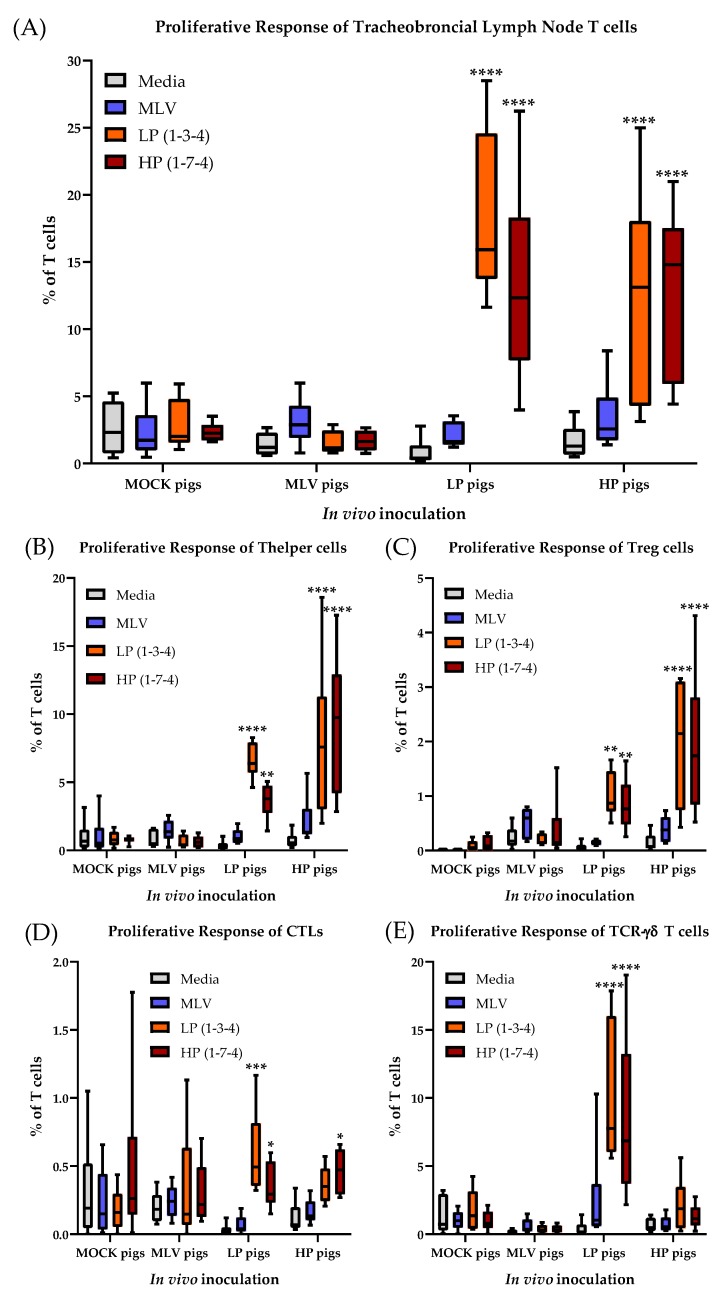
Proliferation of tracheobronchial lymph node T-cells is comparable with 28 dpi. (**A**) T-cell proliferation; (**B**) Th proliferation; (**C**) Treg proliferation; (**D**) CTL proliferation; (**E**) TCR-γδ T cell proliferation. Proliferation was quantified as the percentage proliferating of all T cells or the respective subset. The data were analyzed using Fisher’s LSD repeated-measures 2-way ANOVA comparing all PRRSV-2 restimulations within an in vivo inoculation to media. *n* = 6 for all groups except LP pigs/MLV, (*n* = 5). **** *p* < 0.0001. *** *p* < 0.001, ** *p* < 0.01, * *p* < 0.05.

**Figure 9 viruses-11-00796-f009:**
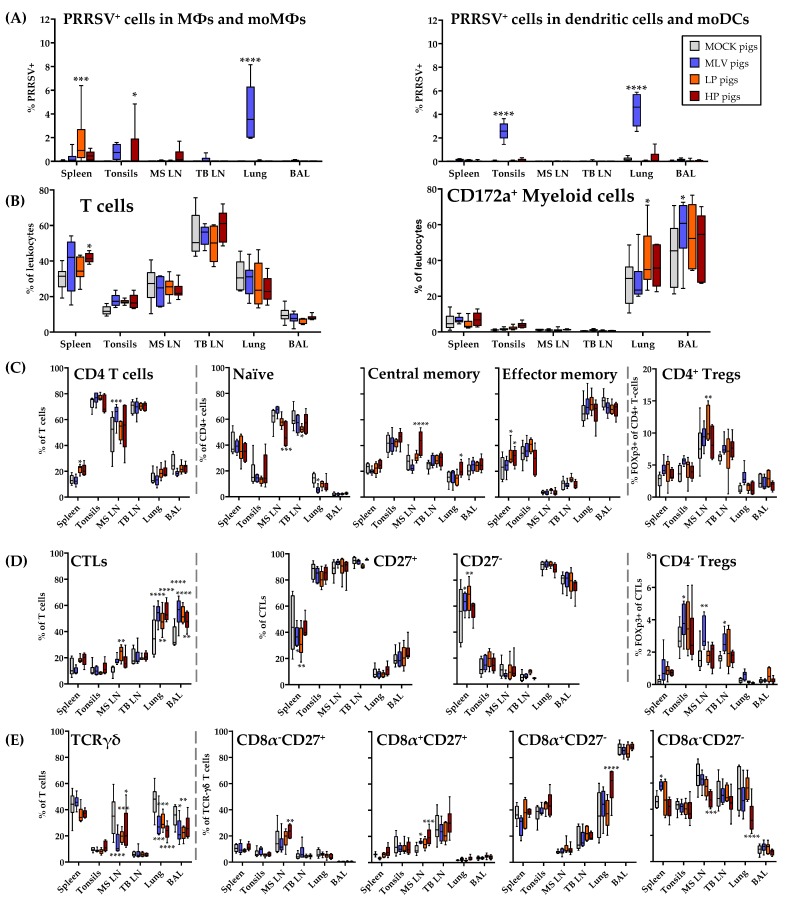
PRRSV-2 inoculation results in increased CTLs at PRRSV-2 routes of entry 63 dpi. (**A**) PRRSV-2^+^ cells in select tissues at 63 dpi; (**B**–**E**) 9-color immunophenotyping staining panel and Treg staining panel of immune cells isolated at necropsy, 63 dpi. The colored boxes (**A**) represent respective in vivo inoculations throughout figure. *n* = 6 for all groups. **** *p* < 0.0001. *** *p* < 0.001, ** *p* < 0.01, * *p* < 0.05.

**Figure 10 viruses-11-00796-f010:**
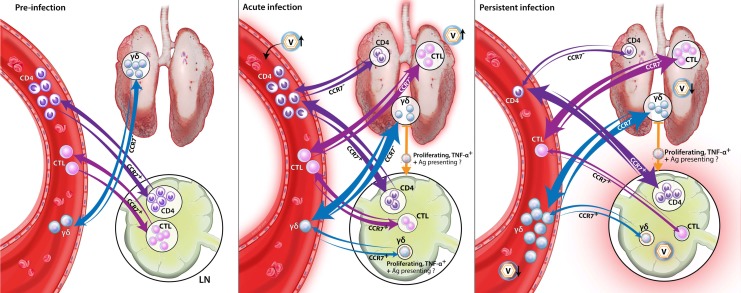
The time course of the T-cell response to the PRRSV-2 infection. Three critical immunological compartments are depicted: blood, lung and tracheobronchial lymph nodes (TB LN); the relative proportions for each are determined as followed: For pre-infection, the data are based on the T-cell frequencies at 0 dpi and from Talker et al. [24]; for acute infection, the relations are based on proliferating cells and CCR7 expression at 28 dpi; for persistent infection, frequencies of proliferating cells at 56 dpi and immune cell staining in tissue at 63 dpi were used. CCR7 expression on immune cells allows for their entrance from the blood into the lymphatic system. In contrast, CCR7^−^ cells circulate between the blood and tissue including the lung. The relative size of the CCR7 arrows indicates the relative proportion of each T-cell subset traveling to each compartment at the indicated states of infection—pre infection, acute infection, and persistent infection. **Pre-infection:** With the exception of cross-reactive or maternally derived T cells, PRRSV-reactive T cells are naïve prior to their first contact with PRRSV. Those PRRSV-2-reactive naïve T cells are circulating in the peripheral blood. Naïve TCR-γδ T cells are CCR7^−^; and they are thereby also located in the lung. In contrast, naïve CD4^+^ T cells and CTLs are CCR7^+^; this CCR7 expression causes them to circulate between blood and the TB LN. **Acute infection:** Upon respiratory PRRSV infection, PRRSV particles first enter the lung and later on also the blood stream (V↑). Therefore, PRRSV-reactive naïve TCR-γδ T cells in the lung can directly encounter the virus. Here they can quickly react to the infection. The majority of those cells can stay CCR7^−^, including IFN-γ-producing TCR-γδ T cells; these cells can migrate between the lung and the blood. However, some of those TCR-γδ T cells can upregulate CCR7 and acquire the ability to home to the lymphatic system. These lymph node homing TCR-γδ T cells can proliferate and/ or produce TNF-α. The function of those cells is currently unknown; but they might induce inflammation and/ or present PRRSV antigen in the lymph nodes to facilitate the clearance of PRRSV from the lymphatic tissues. Naïve TCR-αβ T cells are in the blood and TB LN; antigen presenting cells need to pick-up the antigen, mature, migrate to the lymph node and then present the PRRSV antigen to those naïve TCR-αβ T cells to induce TCR-αβ T cell activation. Once activated, those naïve TCR-αβ T cells differentiate into antigen-experienced T cells and potentially memory cells. The activated CD4^+^ T cells react with strong proliferation leading to a marked increase of antigen-experienced CD8α^+^CD4^+^ T cells. These CD8α^+^ CD4 T cells can be CCR7 negative or positive: Therefore, they can migrate to both, the lung and TB LN, but they slightly prefer the TB LN. Probably based on the increase of CD4 T cells, the relative frequency of PRRSV-reactive CTLs decreases in the blood. The PRRSV-reactive CTLs in the blood are also able to home to the lung and the TB LN. However, the CCR7 expression indicates that these PRRSV-reactive CTLs are slightly more likely to travel to the lung than to the TB LN. **Persistent infection:** During persistent infection, PRRSV is mainly cleared from the lung and the blood (V↓), but PRRSV is still present in lymphoid tissues (V). Many of the T-cell homing characteristics of the acute infection remain or are intensified during persistent infection. The TCR-γδ T cells now comprise the vast majority of the PRRSV-reactive T cells in the blood. While the majority of TCR-γδ T cells still home to the lung, the numerous proliferating TCR-γδ T cells are now also homing to the lymphatic system, including the TB LN. Thereby, these TCR-γδ T cells might play a more prominent role in combating PRRSV infections in the lymphatic system. CD4^+^ T cells are at this late stage confined primarily in the TB LNs. Finally, CTLs have taken up residence in the lungs; they might therefore be relevant for long-term protection at the most likely site for future PRRSV-2 exposures.

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
