# Peer review of "The T-Cell Response to Type 2 Porcine Reproductive and Respiratory Syndrome Virus (PRRSV)"

_viruses, 2019, doi:10.3390/v11090796_

Round 1
Reviewer 1 Report
In this manuscript, the authors provided a valuable information of the immune response in PRRSV infection. I believe this paper article provided important information for pathogenesis, vaccine development and control of PRRSV in the future. Major comments: 1. The reviewer would like to know the ELISA and SN titer of the different PRRSV infection (MLV, HP, LP group) from dpi 0 to 63. As we know CD4+ T cells (T helper cells) also regulate humoral immunity. 2. This a great research finding, I hope the authors could summarized in Figures for temporal sequence of immune response events after infection of different PRRSV strains.Author Response
Suggestions from Reviewer #1:
In this manuscript, the authors provided a valuable information of the immune response in PRRSV infection. I believe this paper article provided important information for pathogenesis, vaccine development and control of PRRSV in the future.
Major comments:
1. The reviewer would like to know the ELISA and SN titer of the different PRRSV infection (MLV, HP, LP group) from dpi 0 to 63. As we know CD4+ T cells (T helper cells) also regulate humoral immunity. We added a summary paragraph describing the results. We plan to submit a separate manuscript describing the humoral immune response. Adding the quite extensive humoral immune response data to this manuscript would have exceeded the realm of this manuscript (Lines 717-722).
2. This a great research finding, I hope the authors could summarized in Figures for temporal sequence of immune response events after infection of different PRRSV strains. We included an illustrated summary figure (Figure 10) and caption to better describe the temporal sequence. (Lines 739-779)
Reviewer 2 Report
The manuscript „The T-cell response to type 2 porcine reproductive and respiratory syndrome virus (PRRSV)“ by Kick et al. provides an in-depth analysis of the porcine T-cell response to PRRSV-2 infection.
Kick et al. investigated the homologous and heterologous response of blood-derived Th cells, Tregs, CTLs, and TCR-gd T cells in the time course following intramuscular injection of a MLV and following intranasal infection with low- and high-pathogenic PRRSV-2 strains. The PRRSV-specific response was assessed by flow cytometric analysis of proliferation and cytokine production (IFN-g, TNF-a, IL-2) combined with analysis of differentiation markers following in vitro restimulation.
At necropsy, 63 days post infection, the PRRSV-specific response was also assessed in tracheobronchial lymph nodes. Moreover, T cells and myeloid cells from spleen, tonsils, mediastinal lymph nodes, tracheobronchial lymph nodes, lung and BAL were flow-analyzed for content of PRRSV and T-cell subsets were analyzed for frequency and differentiation status. This detailed analysis of the T-cell response was accompanied by qPCR of serum samples to detect viremia. Throughout the experiment, pigs were also monitored for clinical signs and changes in body weight and body temperature.
The detailed flow cytometric analyses performed in this study provide new insights into the T-cell response to PRRSV-2 and provide a valuable basis for future work, especially in regard to correlates of protection. Also, the present study introduces a potential role of TCR-gd T cells in a later phase of PRRSV infection, which will be interesting to follow up in future studies. Given that differences in gating strategies might lead to different results, it would be highly valuable to provide the raw data alongside the manuscript.
Main comments:
1) In the gating strategy for the 9-color staining of necropsy tissue (Supplemental Figure 1), CD8a negative cells are included in the gate for TEM. Should it not be CD8a+CD27- only?
2) What is the presumed identity of CD172a++ and CD172a+SSC++ cells? (Gating shown in Supplemental Figure 1). How do these subsets relate to the myeloid subsets gated in Supplemental Figure 3?
3) The 9-color gating shown in supplemental figure 1 lacks doublet discrimination, which is critical especially for the myeloid subsets.
4) Did the authors include FMO controls to ensure proper gating for Foxp3 expressing cells?
5) Doublet discrimination is also lacking in Supplementary Figure 3. Inclusion of doublets could give false-positive events for PRRSV+ cells.
6) Is the gating strategy for monocyte-derived macrophages, monocyte-derived dendritic cells, macrophages and cDC2 (as shown in Supplemental Figure 3) established in porcine immunology? The authors should cite papers that support this gating strategy.
7) Is there a possibility to check whether the rebound in viremia detected at day 56 is due to infectious virus or rather due to viral RNA potentially released by dying cells /macrophages?
8) Figure 7 shows the CCR7 expression on proliferating vs non-proliferating TCR-gd T cells. In order to show that PRRSV-2 increases the expression of CRR7 on TCR-gd T cells (as stated in the heading of figure 7), it would be more relevant to compare with proliferating TCR-gd T cells from Mock-animals (and potentially also infected animals) that were incubated with Medium. The observation that CCR7 expression is lower on non-proliferating TCR-gd T cells does not proof that PRRSV “induces CCR7 on TCR-gd T cells”, as the upregulation may simply be related to proliferation.
9) Apparently, also the first contact with PRRSV (during in vitro restimulation) is able to induce CCR7 on TCR-gd T cells, as seen for the Mock-pigs/MLV in Figure 7 – this should be made clear in the text.
10) How do the authors explain the discrepancy between IFN-g producing and proliferating TCR-gd T cells in regard to their CCR7 expression? The authors hypothesize that activated TCR-gd T cells upregulate CCR7, but why is the majority of IFN-g producing TCR-gd T cells negative for CCR7? The lack of CD8 expression on IFN-g producing TCR-gd T cells is also unexpected and would suggest that IFN-g is produced by naïve cells.
Minor comments:
11) Line 585: replace “are” by “is”
12) Line 270 needs rephrasing – “differentiation into naïve cells” does not make sense
13) Line 252-253: “but the degree of response various to different strains” needs rephrasing
14) Line 380: What do the authors mean by “tissue-draining” CCR7- CTLs? “Tissue-homing” may be more appropriate here.
15) Line 470: For consistency with the other headings, the authors may consider writing “63dpi” instead of “9 weeks post infection”
16) Line 521-522: What does “besides the lymphoid tissues” mean in this context?
17) Lines 681, 682, and 684: delete “%”
Comments on figures:
18) On the printout it is hardly possible to distinguish the shades of red used for LP and HP. The color “orange” could be used instead of "light red". Replacing "green" by "blue" would also account for color-blindness.
19) Figure 1A is hardly readable due to poor contrast (especially printout) in the green box. Writing the text in “black” instead of “dark-green” would improve readability.
20) A darker shade of grey would improve readability in Figure 1 B-E both on the screen and the printout.
21) In the legend of Figure 4A, the symbol for HP pig needs to be inverted.
22) In legend of figure 1B, please clarify the meaning of “a” and “b”
23) In legend of figure 3, 5 and 6, please clarify the meaning of “a” and “b”
24) In general, the resolution of figures is quite low in the provided PDF and should be improved for the final manuscript
Author Response
Suggestions from Reviewer #2:
The manuscript “The T-cell response to type 2 porcine reproductive and respiratory syndrome virus (PRRSV)“ by Kick et al. provides an in-depth analysis of the porcine T-cell response to PRRSV-2 infection.
Kick et al. investigated the homologous and heterologous response of blood- derived Th cells, Tregs, CTLs, and TCR-gd T cells in the time course following intramuscular injection of a MLV and following intranasal infection with low- and high-pathogenic PRRSV-2 strains. The PRRSV-specific response was assessed by flow cytometric analysis of proliferation and
cytokine production (IFN-g, TNF-a, IL-2) combined with analysis of differentiation markers following in vitro restimulation. At necropsy, 63 days post infection, the PRRSV-specific response was also assessed in tracheobronchial lymph nodes. Moreover, T cells and myeloid cells from spleen, tonsils, mediastinal lymph nodes, tracheobronchial lymph
nodes, lung and BAL were flow-analyzed for content of PRRSV and T-cell subsets were analyzed for frequency and differentiation status. This detailed analysis of the T-cell response was accompanied by qPCR of serum samples to detect viremia. Throughout the experiment, pigs were also monitored for clinical signs and changes in body weight and body
temperature. The detailed flow cytometric analyses performed in this study provide new insights into the T-cell response to PRRSV-2 and provide a valuable basis for future work, especially in regard to correlates of protection. Also, the present study introduces a potential role of TCR-gd T cells in a later phase of PRRSV infection, which will be interesting to follow up in future studies. Given that differences in gating strategies might lead to different results, it would be highly valuable to provide the raw data alongside the manuscript.
The raw data include results which are the basis for currently ongoing and future studies of our lab, including the fore-mentioned study on the B-cell and humoral immune response (see replies to reviewer 1). Therefore, we would prefer not to provide these raw data at this point yet.
Main comments:
1) In the gating strategy for the 9-color staining of necropsy tissue (Supplemental Figure 1), CD8a negative cells are included in the gate for TEM. Should it not be CD8a+CD27- only? It has been shown in humans that TEM can downregulate their memory marker when they become terminally differentiated. In addition, recent data from the Saalmüller lab indicate that subsets of multifunctional T cells are CD8a-CD27-. These yet unpublished results were presented at the IVIS 2019 in Seattle. Therefore, we assume that the CD8adimCD27- cells are on their way to become those CD8a-CD27- cells; but the separation in our case is not clear enough to make them a separate subset. Therefore, we included the CD8adimCD27- cells in the TEM fraction.
2) What is the presumed identity of CD172a++ and CD172a+SSC++ cells? (Gating shown in Supplemental Figure 1). How do these subsets relate to the myeloid subsets gated in Supplemental Figure 3? It is not possible with only CD172a and SSC to identify myeloid subsets and we did not stain for other myeloid markers in this 9-color panel; we therefore changed the gate to include the complete CD172a+ myeloid population.
3) The 9-color gating shown in supplemental figure 1 lacks doublet discrimination, which is critical especially for the myeloid subsets. We added doublet discrimination. Within a mixed cell population, doublet discrimination should be used with caution since the different cell types will display different H, W, and A properties. We therefore used a generous doublet discrimination gate to prevent the exclusion of less-frequent immune subsets such as myeloid cells.
4) Did the authors include FMO controls to ensure proper gating for Foxp3 expressing cells? Yes. We added this information to the text (lines 153, 162, 175).
5) Doublet discrimination is also lacking in Supplementary Figure 3. Inclusion of doublets could give false-positive events for PRRSV+ cells. We added doublet discrimination. As explained above, we used a rather generous gate.
6) Is the gating strategy for monocyte-derived macrophages, monocyte-derived dendritic cells, macrophages and cDC2 (as shown in Supplemental Figure 3) established in porcine immunology? The authors should cite papers that support this gating strategy. Yes, we added the reference for the gating strategy. (line 151, reference #23)
7) Is there a possibility to check whether the rebound in viremia detected at day 56 is due to infectious virus or rather due to viral RNA potentially released by dying cells /macrophages? We agree with the reviewer that additional tests would be beneficial to answer this question. This test would require a TCID50 determination within sera of this animal trial. Unfortunately, we utilized our limited serum for other analyses on various aspects of the humoral response. Therefore, we do not have any remaining serum to perform this test. While we are unfortunately not able to provide those TCID50 data, we will definitely include this test in future studies.
8) Figure 7 shows the CCR7 expression on proliferating vs non-proliferating TCR-gd T cells. In order to show that PRRSV-2 increases the expression of CRR7 on TCR-gd T cells (as stated in the heading of figure 7), it would be more relevant to compare with proliferating TCR-gd T cells from Mock- animals (and potentially also infected animals) that were incubated with Medium. The observation that CCR7 expression is lower on non-proliferating TCR-gd T cells does not proof that PRRSV “induces CCR7 on TCR-gd T cells”, as the upregulation may simply be related to proliferation. The reviewer is correct that our statement is misleading. We edited this statement on our results to reflect your comments. (lines 441-445)
9) Apparently, also the first contact with PRRSV (during in vitro restimulation) is able to induce CCR7 on TCR-gd T cells, as seen for the Mock-pigs/MLV in Figure 7 – this should be made clear in the text. Same as #8 (lines 441-445)
10) How do the authors explain the discrepancy between IFN-g producing and proliferating TCR-gd T cells in regard to their CCR7 expression? The authors hypothesize that activated TCR-gd T cells upregulate CCR7, but why is the majority of IFN-g producing TCR-gd T cells negative for CCR7? The lack of CD8 expression on IFN-g producing TCR-gd T cells is also unexpected and would suggest that IFN-g is produced by naïve cells. We thank the reviewer for this excellent comment. We explain this discrepancy by the different roles TCR-gd T cells can play. We assume that TCR-gd T cells in the periphery produce IFN-g to the anti-viral response by macrophage activation; in contrast, TCR-gd T cells in the lymph nodes rather produce other cytokines such as TNFa (see Fig. 6c) In addition, we hypothesize that TCR-gd T cells homing to the LN play a part in antigen presentation. Further research on the role of those TCR-gd T cells in the different tissues will be required to answer those questions. We edited the discussion to include this apparent discrepancy. (lines 637-650 + 665)
Minor comments:
11) Line 585: replace “are” by “is”. Completed (line 586)
12) Line 270 needs rephrasing – “differentiation into naïve cells” does not make sense. Reworded (line 274)
13) Line 252-253: “but the degree of response various to different strains” needs rephrasing. Reworded (line 257)
14) Line 380: What do the authors mean by “tissue-draining” CCR7- CTLs? “Tissue-homing” may be more appropriate here. Changed (line 378)
15) Line 470: For consistency with the other headings, the authors may consider writing “63dpi” instead of “9 weeks post infection” Updated (line 474) and also updated Figure 9 caption (lines 543-545)
16) Line 521-522: What does “besides the lymphoid tissues” mean in this context? Reworded (line 521-522)
17) Lines 681, 682, and 684: delete “%”. Deleted the %. (lines 694-697)
Comments on figures:
18) On the printout it is hardly possible to distinguish the shades of red used for LP and HP. The color “orange” could be used instead of "light red". Replacing "green" by "blue" would also account for color-blindness. We made the suggested changes to the color scheme for all figures and as described in the text. (lines 118, 119, 195)
19) Figure 1A is hardly readable due to poor contrast (especially printout) in the green box. Writing the text in “black” instead of “dark-green” would improve readability. We changed the color to “black”.
20) A darker shade of grey would improve readability in Figure 1 B-E both on the screen and the printout. We changed to a darker grey.
21) In the legend of Figure 4A, the symbol for HP pig needs to be inverted. We inverted the symbol.
22) In legend of figure 1B, please clarify the meaning of “a” and “b”. We reworded the description for the meaning of “a” and “b”.
23) In legend of figure 3, 5 and 6, please clarify the meaning of “a” and “b”. We reworded the description for the meaning of “a” and “b”.
24) In general, the resolution of figures is quite low in the provided PDF should be improved for the final manuscript. We inserted a higher quality .png into the manuscript. The figure .pdfs are to the required RGB quality.